# Limited impact of El Niño – Southern Oscillation on variability and growth rate of atmospheric methane

Hinrich Schaefer[1], Dan Smale[1], Sylvia E. Nichol[1], Tony M. Bromley[1], Gordon W. Brailsford[1], Ross J. Martin[1], Rowena Moss[1], Sylvia Englund Michel[2], James W. C. White[2]

[1]Climate and Atmosphere Centre, National Institute of Water and Atmospheric Research, Wellington, 6021, New Zealand
[2] Institute of Arctic and Alpine Research, Boulder, CO, USA.

*Correspondence to*: Hinrich Schaefer (Hinrich.Schaefer@niwa.co.nz)

**2. Abstract.** The El Niño – Southern Oscillation (ENSO) has been suggested as a strong forcing in the methane cycle and as a driver of recent trends in global atmospheric methane mole fractions [$CH_4$]. Such a sensitivity of the global $CH_4$ budget to climate events would have important repercussions for climate change mitigation strategies and the accuracy of projections for future greenhouse forcing. Here, we test the impact of ENSO on atmospheric $CH_4$ in a correlation analysis. We use local and global records of [$CH_4$], as well as stable carbon isotopic records of atmospheric $CH_4$ ($\delta^{13}CH_4$), which are particularly sensitive to the combined ENSO effects on $CH_4$ production from wetlands and biomass burning. We use a variety of nominal, smoothed and detrended time series including growth rate records. We find that at most 36% of the variability in [$CH_4$] and $\delta^{13}CH_4$ is attributable to ENSO, but only for detrended records in the Southern tropics. Trend-bearing records from the Southern tropics, as well as all studied hemispheric and global records show a minor impact of ENSO, i.e. <24% of variability explained. Additional analyses using hydrogen cyanide (HCN) records show a detectable ENSO influence on biomass burning (up to 51%-55%), suggesting that it is wetland $CH_4$ production that responds less to ENSO than previously suggested. Dynamics of the removal by hydroxyl likely counteract the variation in emissions, but the expected isotope signal is not evident. It is possible that other processes obscure the ENSO signal, which itself indicates a minor influence of the latter on global $CH_4$ emissions. Trends like the recent rise in atmospheric [$CH_4$] can therefore not be attributed to ENSO. This leaves anthropogenic methane sources as the likely driver, which must be mitigated to reduce anthropogenic climate change.

3. Keywords:  Time series analyses, isotopes, climate variability, climate change, methane, stable isotope analysis [more]

## 4. Introduction

Attributing recent changes in the methane budget, and the associated impact on its growth rate, to specific natural or anthropogenic causes is essential for climate change mitigation. The impact of climatic variability on methane emissions is particularly important to assess the potential for $CH_4$ release under future climate scenarios (e.g., from permafrost and wetland environments, as well as gas hydrates) in a reinforcing feedback. Atmospheric methane mole fractions [$CH_4$] have increased by 140% over preindustrial levels (MacFarling Meure et al., 2006). The associated increase in radiative forcing makes $CH_4$ the second-most important anthropogenic greenhouse gas (Shindell et al., 2009). The long-term [$CH_4$] increase until the late 1990s can be attributed to increasing emissions from fossil fuel production (Ferretti et al., 2005; Schaefer et al., 2016), as well as sources from agriculture (enteric fermentation in livestock, rice production), waste management and anthropogenic burning (van Aardenne et al., 2001; Saunois et al., 2016). After a plateau in the early 2000s, [$CH_4$] has been rising again since 2007. Considering recent reconstructions of methane's dominant atmospheric sink, i.e. the hydroxyl radical OH, we consider it likely that increasing emissions contribute to (Rigby et al., 2017), if not dominate (Naus et al., 2018), the [$CH_4$] rise. If so, the methane source type that varied can be investigated with measurements of stable carbon isotope ratios in atmospheric methane ($\delta^{13}CH_4$). The latter are influenced by the relative source contributions from $^{13}C$-depleted biogenic, $^{13}C$-rich pyrogenic, and thermogenic methane with intermediate $\delta^{13}C$. Isotope studies suggest that biogenic methane sources make either a dominant (Schaefer et al., 2016; Nisbet et al., 2016) or strong (Worden et al., 2017) contribution to the recent [$CH_4$] rise. Biogenic methane comes predominantly from wetlands and agriculture. Schaefer et al, (2016) suggested agriculture as the more likely cause, primarily because satellite data place the increased emissions in Southeast Asia, India and China (Houweling et al., 2014). However, this geographic footprint from an inversion of satellite data is also consistent with fluxes from one particular wetland emissions model (Houweling et al., 2014). Other studies also assume a stronger role of wetlands due to drier conditions during the plateau years (Bousquet et al., 2006) and higher wetland emissions afterwards, which are attributed to a switch to predominant La Niña conditions around 2007 (Bousquet et al., 2011; Nisbet et al., 2016). La Niña is the cold phase of El Niño – Southern Oscillation (ENSO) cycles, which have a strong impact on precipitation anomalies in tropical regions (Ropelewski and Halpert, 1987; Lyon and Barnston, 2005) (Fig.1) that are key source areas for methane production from wetlands and biomass burning (Kirschke et al., 2013). ENSO impacts are strongest in the tropics, generally from December to February. During El Niño (La Niña) events in the December to February period, it tends to be drier (wetter) in the Indonesian region, north-east Brazil and south-eastern Africa, whereas it tends to be wetter (drier) in the southern USA and Mexico, eastern China and Taiwan, and east-central Africa (Fig. 1). During El Niño (La Niña) events in the June to August period, it tends to be drier (wetter) in the Indonesian region, central America and India.

The generally drier conditions during El Niños suppress global wetland emissions in models by up to 19 Tg/yr in the 1990s (Hodson et al., 2011). Several anthropogenic sources are subject to the same ENSO forcing and are expected to vary in concert with wetlands (e.g., rice agriculture, possibly livestock). At the same time, dry El Niño phases enhance $CH_4$ emissions from both natural and anthropogenic biomass burning (van der Werf et al., 2006). Wet La Niña conditions have the opposite effect;

summed across the globe they increase wetland emissions and lower biomass burning $CH_4$. As tropical wetland fluxes are considerably larger than biomass burning emissions (Saunois et al., 2017), the expected net effect is a lower [$CH_4$] growth rate caused by El Niño conditions and a higher one due to La Niñas. The ENSO impact on $\delta^{13}CH_4$ should be more pronounced than the one on [$CH_4$], because changes in wetland and biomass burning emissions combine to enrich atmospheric $CH_4$ in $^{13}C$

during El Niños and deplete it during La Niñas. Biogenic methanogenesis in wetlands discriminates strongly against $^{13}C$ and creates methane that is $^{13}C$-depleted ($\delta^{13}C = -58‰$ for tropical wetlands) relative to the plant precursor material ($\delta^{13}C$ of -12‰ to -28‰) and to the combined total of global emissions ($\delta^{13}C \sim -53.5‰$). In contrast, during burning the isotope ratios of the precursor plant material are essentially conserved and lead to $\delta^{13}C \sim -22‰$ for $CH_4$ emissions from fires (Schwietzke et al., 2016). The simultaneous suppression of $^{13}C$-depleted wetland $CH_4$ and enhancement of very $^{13}C$-rich pyrogenic emissions

(and vice versa) act in the same direction on the $\delta^{13}CH_4$ of the combined source. The latter should be detectable in atmospheric $\delta^{13}CH_4$ records if the impact of ENSO on the $CH_4$ cycle is sufficiently large, as is predicted by the emission anomalies in wetland emission models (Hodson et al., 2011), reconstructed from satellite observations of burned area (van der Werf et al., 2010), and observed through variability in hydrogen cyanide (HCN) (Pumphrey et al., 2018), which is an indicator of biomass burning.

Varying contributions from wetlands dominated by $C_3$ and $C_4$ plants, which differ in the $\delta^{13}CH_4$ of their emissions, may be part of the ENSO-$CH_4$ signal or work to obscure it if controlled by other drivers. In general, we assume that $\delta^{13}CH_4$ of the various emission sources has not changed over the ~35 yr period of our study. Although such changes, correlated to atmospheric $CO_2$ mole fractions, have been reported to occur over centuries to millennia in ice core studies (Möller et al., 2013), they are likely negligible over the short duration and >20% $CO_2$-change of our study period.

Changes in OH have also been suggested as partial or dominant drivers in recent $CH_4$ trends, both for the onset of the 1999-2006 plateau (McNorton et al., 2016; Schaefer et al., 2016) and for the post-2007 [$CH_4$] increase (Rigby et al., 2017; Turner et al., 2017). A chemistry climate model suggests that ENSO modulates tropical OH (where hydroxyl levels are highest) via changes in NOx production through lightning, ozone availability and specific humidity, as well as emissions of reactive carbon (Turner et al., 2018). Resulting changes in methane removal could create their own signal in atmospheric records of [$CH_4$] and

$\delta^{13}CH_4$. They could also either reinforce or dampen the emission impacts discussed above.

We conduct correlation analyses between ENSO variability and [$CH_4$], as well as $\delta^{13}CH_4$ records to quantify how much ENSO anomalies in emissions and sinks affect atmospheric $CH_4$. Specifically, we explore how much of the year-to-year variability in atmospheric methane can be attributed to ENSO and how large the ENSO-$CH_4$ signal is in dependence of latitude. We test if recent trends in methane growth rate can be attributed to wetland emissions controlled by ENSO dynamics or if agricultural

sources are more likely drivers. ENSO is quantified by four different indices, which are based on ocean temperature, sea level pressure gradients and a multivariate combination. [$CH_4$] and $\delta^{13}CH_4$ time series from four different locations were used, two from stations in the Southern tropics (Samoa, SMO, and Ascension Island, ASC), the Southern mid-latitudes (Baring Head, NZ; BHD) taken as representative of the Southern hemisphere, and global average time series of [$CH_4$] and $\delta^{13}CH_4$ calculated from a network of global stations (Dlugokencky et al., 2011; Schaefer et al., 2016). We also investigate ENSO's impact on

HCN data measured in Lauder, NZ (LAU), to quantify the biomass burning contribution separately. The aim is to detect the impact of ENSO on atmospheric $CH_4$ on various spatial scales.

## 5.1. Methods

### 5.1.1. Data

For access to all data sets used in this study see Sect. 10

#### 5.1.1.1. ENSO indices

We used four different indices in our analysis to cover various climatic effects of the ENSO cycle (Figs. 2A and 3A). The Southern Oscillation Index (SOI) is calculated from the gradient in mean sea-level pressure observations at Tahiti and Darwin, Australia (Troup, 1965). Further information on the SOI is given by (Horel and Wallace, 1981; Trenberth, 1976). The Ocean
Niño Index (ONI) uses sea surface temperature (SST) anomalies in the eastern Pacific Niño 3.4. region (5°N-5°S, 120-170°W), which show smaller intra-seasonal variability than pressure and are further smoothed by using 3-month running means (Barnston et al., 1997; Kousky and Higgins, 2007).

The El Niño Modoki Index (EMI) is based on SST anomalies in the central Pacific (Ashok et al., 2007) rather than the eastern Pacific (the canonical El Niño). Events with the largest SST anomalies in the Modoki region show differences in the climate
teleconnections to canonical El Niño events. The tropical precipitation differences are modest, but large differences in tropospheric circulation and wind anomalies (Yeh et al., 2009) can produce large extra-tropical differences in precipitation and temperature. The EMI has also been shown to be a significant predictor of tropical atmospheric ozone variations (Xie et al., 2014).

Variability in both atmospheric pressure and SST anomalies informs the Multivariate ENSO Index (MEI) (Wolter and
Timlin, 1993; Wolter and Timlin, 1998). The various indices correlate highly with each other ($r^2 = 0.85$ and higher), except the EMI ($r^2$ between 0.33 and 0.52 for correlations with SOI, ONI and MEI), which deviates from the others during the strong 1997-198 El Niño event. Excluding the latter brings the correlation to $r^2$ between 0.74 and 0.79.

An ENSO index based on precipitation data, the ESPI, (Curtis and Adler, 2000) correlates very highly with the MEI, the ONI, and the SOI ($r^2$ of 0.902, 0.909, and 0.839, respectively). Therefore, we did not conduct separate calculations for the
ESPI.

### 5.1.1.2. [CH4] time series

The [CH4] time series used in this study are from the Global Monitoring Division of the National Oceanic and Atmospheric Administration - Earth System Research Laboratory (NOAA ESRL) Carbon Cycle Cooperative Global Air Sampling Network (Dlugokencky et al., 2017). These data include records from SMO (latitude $14.24^0$S, longitude $170.57^0$W) and ASC ($7.92^0$S,
$14.42^0$W) as well as global averages calculated by smoothing background data temporally and zonally; all with coverage from 1983-2017. In addition, we use data measured at the NZ National Institute of Water and Atmospheric Research (NIWA) from BHD in NZ ($41.41^0$S, $174.87^0$E; 1992-2017) (Lowe et al., 1991). Both data sets are on the same international scale (Dlugokencky et al., 2005), although for the presented analysis internal consistency of the time series is the relevant criterion; inter-laboratory offsets do not affect the findings. The individual time series (Figs. 2B-E) show seasonal cycles, inter-annual
variability (IAV) and long-term trends. To investigate ENSO effects on these different time scales we derived the following seven records from the individual measurements at each station (Table 1). First, the nominal monthly mean values to capture the full variability in the data ("nom"). Second, 12-month running means to represent IAV and trends ("run"). Third, monthly resolved growth rate defined as the difference between the following 12 months and the preceding 12 months ("gro"). Fourth, a residual ("res") as calculated by seasonal trend analysis by Loess (STL, Cleveland et al., 1990). The seasonal window was
set at 120 months, which forces a uniform seasonal cycle over the duration of the record. The residual therefore represents IAV in the expression of the seasonal cycle as well as other short-term anomalies. Fifth, sixth, and seventh: detrended time series where the STL trend component is subtracted from the monthly means with subsequent determination of detrended monthly means, 12-month running means, and growth rate ("det-nom", "det-run", "det-gro").

### 5.1.1.3. $\delta^{13}CH_4$ time series

The $\delta^{13}CH_4$ time series used in this study were measured at three different laboratories, i.e., the Institute of Arctic and Alpine Research (INSTAAR), USA; the Institute for Environmental Physics (IUP) at Heidelberg University, Germany; and at NIWA. Details of the analytical methods are given by Schaefer et al. (2016) and references therein. All values are based on measured $^{13}C /^{12}C$ ratios and are reported in the standard δ-notation $\delta^{13}C = (R_{sample}/R_{standard}-1)*1000‰$ as per mille (‰) values where the reference standard is Vienna PeeDee Belemnite. Records at SMO (1998-2016) and ASC (2000-2016) are measured at
INSTAAR. The BHD record (1992-2016) is based on measurements at INSTAAR and NIWA. An annually averaged global $\delta^{13}CH_4$ time series was established by Schaefer et al. (2016) based on data from INSTAAR, NIWA and IUP. In this analysis, we use the measurements covering 1992-2016 (Fig. 3C). For the global $\delta^{13}CH_4$ data set we conducted the analysis for the nominal annual means ("nom") and growth rate, i.e. difference between two subsequent yearly values ("gro"). We also detrended the time series by subtracting linear trends for the sub-periods 1992-1999 and 2007-2016 ("det") and then calculated
a detrended growth rate ("det-gro"). For the single-station $\delta^{13}CH_4$ records of BHD, ASC, and SMO we derived the same seven records as described for the [CH4] data (Fig. 3D-F).

### 5.1.1.4. HCN time series

HCN retrievals were computed from mid-infrared solar spectra measured at LAU (45.04°S, 169.68°E) as part of the Network for the Detection for Atmospheric Composition Change (NDACC). The time series has been described by Zeng et al. (2012), but the data used here are from updated retrievals using the improved SFIT4 algorithm (NDACC, 2014). The HCN data show strong seasonality that is even more pronounced in the updated retrievals. Zeng et al. (2012) found a significant negative trend for 1997-2009 and attributed it to variations in biomass burning. A similar deseasonalised trend is apparent in the updated record. HCN values are here reported as Petamolecules cm$^{-2}$. Measurements cover the period 1998-2017 when combined for two different instruments with a change-over point in 2000. We conducted our analyses for total column values (0-100 km). The latter signal is dominated by the tropospheric burden as measured in the 0-12 km height partial column; the correlation between total and tropospheric HCN is $r^2 = 0.997$. In addition, we investigated whether the stratospheric HCN burden is differently impacted by ENSO. To that end, we used the 12-100 km partial column, which holds ~22% of the total HCN burden. This layer shows lower correlation with the total column record ($r^2 = 0.45$).

Analogous to the monthly resolved methane records we constructed monthly means, 12-month running means, growth rates and STL residuals for the total column and stratospheric HCN data (Fig. 3B). No detrended records other than STL residual were considered.

### 5.1.2. Analysis

We conducted correlation analyses between the time series of a chosen ENSO index and either a [CH$_4$], $\delta^{13}$CH$_4$, or HCN record as the dependent variable. The degree of correlation is quantified by the square ($r^2$-value) of the Pearson correlation coefficient or, alternatively, of the Spearman ranking coefficient. The Pearson coefficient is more commonly used, but it assumes linear relationships between the variables and may underestimate nonlinear correlations. We therefore also used the Spearman rank, which does not require linearity. Note that not all correlation combinations were tested using both coefficients.

A lag time between ENSO forcing and detection of resulting $\delta^{13}$CH$_4$ or HCN variability at the measurement site, (or in the global average) is likely, due to a variety of factors that may lead to lags of unknown length and some of which may be cumulative: e.g., hydrology, plant growth and decay, microbial response, seasonal triggers for methanogenesis or burning, as well as atmospheric chemistry, mixing, and transport between source regions and sampling sites. Therefore, it is difficult to define a cut-off for lags. Literature estimates of specific lags range from days (Chamberlain et al., 2016) to 7 months (Zhang et al., 2018; Zhu et al., 2017), not counting atmospheric transport. Given ENSO variability with a periodicity of 2-7 years (McPhaden et al., 2006), our analysis therefore allows for lag times of up to 5 years in monthly increments in the calculations and reports the maximum $r^2$ and lag time (in months) for a given ENSO-[CH$_4$]/$\delta^{13}$CH$_4$/HCN combination. We conducted the analysis for all permutations of the four ENSO indices as monthly means and their 12-month running means as well as the [CH$_4$], $\delta^{13}$CH$_4$, and HCN data products listed in Sect. 2.1 and Table 1. For all [CH$_4$], $\delta^{13}$CH$_4$, and HCN parameters we used the period 1998-2016, except for ASC where data are available only from late 2000. Using the same period for all time series

avoids differing correlation results due to varying data coverage. The period includes the strong El Niños of 1998 and 2015, as well as the strong La Niñas of 1999, 2007 and 2010. We also calculated correlations for the period 1983-2016 ([CH$_4$] of SMO, ASC and global) and 1992-2016 ($\delta^{13}$CH$_4$ at BHD and global).

## 5.2. Results

Most combinations have r$^2$-values <0.1 when comparing one dependent data set to the different ENSO time series (Tables 2-4). In the following, we only summarise results for the highest r$^2$ for each dependent time series (across all the nominal, smoothed and detrended records for a station). Given that Pearson coefficient and Spearman rank give comparable results (Tables 3 and 4), we quote the Spearman results unless otherwise mentioned. P-values for the Spearman ranks indicate that all results for r$^2$>0.1 are significant (p<0.001), except for global $\delta^{13}$CH$_4$ correlations, where no p-values below 0.05 occur.

Although the analysis provides r$^2$-values for lags up to 60 months (Tables 2-4), we consider it likely that lags of >3 years indicate spurious correlations given that individual ENSO events last 1-2 years and global atmospheric mixing times are on order of 1 year. Therefore, we also report the highest r$^2$ for lags <3 years in the following sections. For other cases with lags >3 years in Tables 2-4, the highest relevant r$^2$-value is lower than the reported value, where the latter places an upper limit on the influence of ENSO.

Methane mole fractions show correlations with ENSO of r$^2$-values up to 0.36 at SMO, but only for detrended time series (Table 1). The highest values are from (detrended) growth rates, which can be more indicative of dynamics within an ENSO event, rather than its overall emissions impact (Zhang et al., 2018). For SMO detrended [CH$_4$] series, lag times are fairly consistent across the various ENSO indices and generally shorter than 1 year. For other [CH$_4$] records at SMO and ASC the highest correlations are r$^2$<0.23 and have lags of over 3 years (r$^2$<0.19 for lags <3 years). The global running mean [CH$_4$] time series

shows r$^2$=0.24 (lag: 4.5 years; r$^2$=0.04 for lag <3 years) with the SOI running mean for the period 1998-2016. However, for the full length of available data, as well as all BHD records, all correlations are below r$^2$=0.20, with lag times that are variable, extremely short (zero or 1 month) or over 3 years.

The highest correlations are between HCN running means for total column, as well as stratospheric growth rates, and 12-month running mean ENSO records (Table 2). Here, ENSO accounts for 30%-51% of the observed variability, depending on the

ENSO index. For both total and stratospheric HCN, lag times for maximum correlation are generally shorter than one year and are consistent (≤6 months difference) between the various ENSO indices, with exception of the EMI.

The $\delta^{13}$CH$_4$ records from the stations SMO, ASC, and BHD all have r$^2$-values below 0.24 (Table 3). Variability in lag times between different ENSO indices for the same dependent record is generally high.

None of the global $\delta^{13}$CH$_4$ series produced statistically robust correlations with ENSO; all p-values were higher than 0.05. The

following findings are therefore not relevant for further interpretation. The highest correlation is between global detrended $\delta^{13}$CH$_4$ and SOI monthly means with r$^2$=0.37. Global $\delta^{13}$CH$_4$, is the only parameter where ENSO monthly means produce higher correlations than the smoothed (12-month running mean) record. Because the correlation calculation between annual

$\delta^{13}CH_4$ and ENSO monthly means is specific for the month-of-year, this indicates that global $\delta^{13}CH_4$ is more sensitive to the seasonality of ENSO than its IAV. The actual ENSO influence on global $\delta^{13}CH_4$ is shown through correlation with running ENSO indices, which is highest between nominal $\delta^{13}CH_4$ values and SOI with Pearson $r^2 = 0.25$ for 1998-2016. For the period 1992-2016 this value drops to Pearson $r^2 = 0.20$. The lack of statistical robustness for global $\delta^{13}CH_4$-ENSO correlations may stem from the different resolution of the two sets of time series. In this case, the southern hemispheric record from BHD may represent the extra-tropical impact of ENSO on $\delta^{13}CH_4$.

The full BHD record for 1992-2016 gives very similar results as the 1998-2016 subset used for comparison with the other stations (as discussed above). However, the shorter subset for 1998-2014 produces larger Pearson $r^2$-values (0.26 for running means and SOI), and for 2001-2014 we find Pearson $r^2$-values up to 0.38 (growth rate correlated to EMI). These shorter data sets omit the strong El Niño events of 1998 and/or 2015-16, which could have been expected to have a strong influence on methane emissions and consequently $\delta^{13}CH_4$.

For none of the stations (including global average) did the detrended $\delta^{13}CH_4$ time series (incl. STL residuals) produce a markedly stronger correlation with ENSO than any of the other data series from that station. This is remarkable because ENSO can be expected to have more influence on IAV than on the long-term trends, which are quite pronounced.

## 5.3. Discussion

5.3.1. General causes and caveats for correlations of [$CH_4$], $\delta^{13}CH_4$, and HCN with ENSO

Detected correlations between ENSO indices and [$CH_4$]/$\delta^{13}CH_4$/HCN by themselves do not prove a causal relationship. However, the underlying mechanisms for a potential forcing have been presented by van der Werf et al. (2006) for biomass burning and by Hodson et al. (2011) for wetland $CH_4$ production. Accordingly, a correlation analysis is useful to quantify an upper limit of variability in the $CH_4$ cycle attributable to ENSO. Because ENSO simultaneously suppresses wetland $CH_4$ that is more [13]C-depleted than the cumulative methane source and enhances pyrogenic $CH_4$ that is more [13]C-enriched (or vice versa), the two influences partly cancel for the combined emission rates, i.e. their impact on [$CH_4$]. However, they reinforce each other's impact on total source $\delta^{13}CH_4$. It is possible that biomass burning and wetland $CH_4$ production have different response times to ENSO forcing, which would weaken their cumulative impact on $\delta^{13}CH_4$. Similarly, longer atmospheric residence time of $CH_4$ (~9 years, Prather et al., 2012) over HCN (~3 months, Li et al., 2000) and a smaller relative portion of ENSO-sensitive emissions in the global methane source may lead to dampening effects in the [$CH_4$] and $\delta^{13}CH_4$ variability and hence lower correlation with ENSO indices compared to HCN. The available records for HCN and $\delta^{13}CH_4$ from ASC and SMO cover only a small number of ENSO events, which could affect the results. However, when analysing subperiods of global and BHD [$CH_4$] and $\delta^{13}CH_4$ records, we find larger correlations for shorter periods, particularly when strong ENSO events are excluded. This shows that the results are not biased against the detection of ENSO influences because records are too short. We also note that all stations measure background air, they are set up to detect broad spatial and temporal trends and

not specific emission events such as an ENSO triggered plume. However, if ENSO is invoked as a main cause of recent trends in [CH$_4$] and $\delta^{13}$CH$_4$ (Nisbet et al., 2016) this should be manifested in sizeable correlations.

### 5.3.2. Contrasting correlation patterns for [CH$_4$] and $\delta^{13}$CH$_4$ versus HCN

In all [CH$_4$] and $\delta^{13}$CH$_4$ records, ENSO cycles explain about one third of the variability in detrended records and less than one quarter in others. This is true even for the Southern tropics, where ENSO has strong climatic impacts and where the majority of low-latitude wetland emissions and of biomass burning emissions originate (Kirschke et al., 2013). Correlations found for ASC and SMO, which represent this latitude band in our study, exceed those for the Southern mid-latitudes or the global record only by a limited margin and only for detrended records. Further, inconsistent lag times, lags of more than three years, and

higher correlation coefficients for the exclusion of major ENSO events point to spurious correlations.

In contrast, we find a prominent influence of ENSO on the biomass burning proxy HCN. ENSO impacts on HCN have been reported before, e.g., by Pumphrey et al. (2018), who observe suppression of HCN levels during La Niña events and enhancement during El Niños, particularly in equatorial Asia. That study found a rather confined geographical impact of El Niño events with strongly enhanced HCN emissions around Malaysia, Indonesia, and Papua New Guinea, as well as generally

rapid transport eastward and to the stratosphere. We speculate that the fast, upward transport (although not observed for all El Niño events) explains why stratosphere growth rates are the most sensitive data set to ENSO. For the total column, the HCN burden is concentrated in lower tropospheric levels and may be subjected to more mixing of different air parcels. According to the results of Pumphrey et al. (2018), data from LAU in the Southern mid-latitudes are outside the region of the strongest HCN signal. This is also evident in the zonal mean HCN climatologies of Sheese et al. (2017). Yet, ENSO accounts for up to

51% of the variability in our biomass burning proxy record. One explanation for the lower combined wetland-pyrogenic $\delta^{13}$CH$_4$ signal is low sensitivity of wetland CH$_4$ production to ENSO events. This is consistent with r$^2$-values of 0.12-0.26 between modelled wetland methane emissions (using different climate data sets as drivers) and MEI as reported by Zhang et al. (2018). Alternatively, other processes in the CH$_4$-cycle obscure the ENSO impacts.

### 5.3.3. Impact of ENSO on methane emission rates

In a correlation analysis by Zhu et al. (2017), ENSO explained 49% of IAV in modelled tropical wetland CH$_4$ emissions. This is far higher than the combined effect with biomass burning on $\delta^{13}$CH$_4$ in this study and therefore seems to be an overestimate. Even so, the magnitude of the modelled emission changes is 6 Tg/yr at most. The modelling study of Hodson et al. (2011) finds slightly larger anomalies in global wetland emissions due to ENSO with mean reductions of -9±3 Tg/yr and

mean gains of +8±4 Tg/yr for El Niño and La Niña events, respectively. Pandey et al. (2017) found in a comprehensive inversion study that the net effect of the strong 2011 La Niña on tropical and northern extratropical CH$_4$ emissions was a global increase of +6.6 Tg/yr. The wetland emission anomalies are expected to be partly compensated by changes in biomass burning that are of opposite sign. We are not aware of studies that quantify biomass burning anomalies for specific ENSO events. Assuming that ENSO is the main control of biomass burning emissions of CH$_4$, the IAV in the GFED data (van der

Werf et al., 2010) may serve as an indication for possible ENSO impacts. In that case, the standard deviation of 2.4 Tg/yr for 1998-2014 would approximate the average impact, with maximum anomalies of up to 4 Tg/yr. We use these numbers together with results from Hodson et al. (2011) in the following proof-of-concept discussions. The combined wetland – biomass burning anomalies are ~6 Tg/yr for average ENSO events and ~8 Tg/yr for extreme ones; restricted to 1-2 yearlong individual events. This is well short of the sustained increase after 2007 when yearly emissions were ~20 Tg higher than during the 1999-2006 plateau period and the 9 Tg/yr reduction during the 1990s (Schaefer et al., 2016). Previous findings that modelled tropical (Zhu et al., 2015) and global (Zhang et al., 2018) wetland $CH_4$ emissions can explain at most 25% and 14%, respectively, of the variation in atmospheric methane growth rates therefore agree with our results that ENSO exerts only a minor control on atmospheric $CH_4$.

### 5.3.4. Process based understanding of ENSO impact on wetlands

A major contribution of ENSO to the recent [$CH_4$] increase is inconsistent with independent assessments of wetland response, as shown above, but our findings do not detect any clear minor contribution of ENSO to [$CH_4$] and $\delta^{13}CH_4$ timeseries, either. Several reasons may explain the lack of correlation, where we assume that wetlands respond less than proposed. The main ENSO forcing on tropical wetland $CH_4$ production is thought to be via wetland extent, which is driven by precipitation (Hodson et al., 2011; Holmes et al., 2015; in contrast Zhu et al., 2017, find temperature to be dominant). However, a case study in the Eastern Amazon finds that precipitation changes explain only 21% of wetland $CH_4$ emission variance during the wet season and 7% over the whole year (Basso et al., 2016). The lack of a direct link between precipitation and wetland $CH_4$ production is also evident in the large range in output from various wetland models even when forced with the same meteorological conditions (Melton et al., 2013), although the disagreement between models could also be due to an incomplete understanding of influences on the wetland cycle other than precipitation (Turetsky et al., 2014; Bridgham et al., 2013; Parker et al., 2018). Zhang et al. (2018) report an evolving response of wetland emissions to El Niños, where an initial reduction due to decreased wetland extend is counteracted by increased microbial activity under higher temperatures during the later stages of the event. A complex response of wetland $CH_4$ production is not only seen in models, however. The inversion study of Pandey et al. (2017) found a global increase of +6.6 Tg/yr for the strong 2011 La Niña, but a reduction by -6.1 Tg/yr during the 2012 weak La Niña. Similarly, Liu et al. (2017) found that El Niño conditions produced opposing weather forcing and carbon cycle responses between various tropical regions, as well as differing ones between the 1998 and 2015 events. Another example of this is flooding in the Amazon region during La Niña events, while flooding in the wetlands of the Paraná region occurs during El Niños (Parker et al., 2018). Depending on the strength and geographical expression of the climate anomaly, ENSO may thus cause regional or global emission anomalies that are opposite to the expected pattern.

### 5.3.5. Evaluating the consistency of ENSO impacts throughout the record

The atmospheric [$CH_4$] history shows global emission reductions in the 1990s and increases after 2007 (Schaefer et al., 2016). This would be consistent with ENSO forcing of the methane cycle where the 1990s were dominated by drier El- Niño periods,

whereas the recent years of predominant La Niña conditions were wetter. Given that the magnitude of the low-latitude wetland CH$_4$ source exceeds pyrogenic emissions rates, the expected emissions history would qualitatively match atmospheric trends. Also, for a short period between 2008 and 2011 Schaefer et al. (2016) observed the activation of CH$_4$ emissions with an extremely $^{13}$C-depleted cumulative $\delta^{13}$CH$_4$ (~-75‰). Such a value on the global scale is hard to match by a single source type.

The cumulative effect of wetland enhancement and fire suppression forced by the 2008 La Niña event would provide an excellent explanation. However, the isotopic signal of the emissions reductions in the 1990s should be similar if ENSO forcing was the cause. In contrast, Schaefer et al. (2016) found that the "lost emissions" during that period are quite $^{13}$C-rich and rather indicate a reduction in fossil fuel methane. An alternative interpretation of these isotope trends by Rice et al. (2016) requires simultaneous reductions of pyrogenic and biogenic emissions, which is also inconsistent with the expected ENSO forcing. A

consistent match between ENSO phases and global $\delta^{13}$CH$_4$ is therefore neither evident in the dominant $\delta^{13}$CH$_4$ trends nor in the correlation analysis presented in this study.

5.3.6. Using isotopes to attribute emission changes

The impact of an ENSO emissions "perturbation" (i.e. the combined emissions anomaly of an event) on atmospheric $\delta^{13}$CH$_4$

can be assessed in isotope mass balance calculations according to:

$$S_{total}*\delta_{total} = S_1*\delta_1 + S_2*\delta_{2+} S_3*\delta_3 \qquad (1)$$

Where, for a given source, S and $\delta$ represent emission rate and $\delta^{13}$CH$_4$, respectively (note that for scenarios discussed here S may be negative, i.e., a reduction in emissions). Using generic isotope source signatures for biogenic, fossil fuel and pyrogenic methane emissions from Schwietzke et al. (2016), we find that the average La Niña perturbations proposed in section 4.3. have

an effective $\delta^{13}$CH$_4$ of -79‰, with -83‰ for extreme ones. As expected, the combined isotope leverage of wetland enhancement and fire reductions on the global source is strong, equalling the leverage of much larger source anomalies (20 Tg/yr) with lower $\delta^{13}$CH$_4$ of ~-60‰ after 2007 as calculated by Schaefer et al. (2016). In addition to the assumed 6 Tg/yr ENSO perturbation, another ~14 Tg/yr of emissions with $\delta^{13}$CH$_4$=-52‰ would be necessary to produce the observed [CH$_4$] and $\delta^{13}$CH$_4$ trends. The isotope mass balance then shows that the non-ENSO additional emissions are a roughly equal mix of

fossil fuel and biogenic methane. Noting that the assumption that all years after 2007 experienced average La Niña conditions is unrealistic; these findings therefore show the following three points: (i) ENSO effects alone cannot explain the post-2007 [CH$_4$]-rise. (ii) There was an increase in biogenic sources in addition to ENSO driven wetland anomalies. Other wetland variability may have contributed to the rise (Zhang et al., 2018); given the range in wetland model output (Melton et al., 2014) this stands to be confirmed by ensemble runs. In the absence of boreal emission increases (Sweeney et al., 2016), the only

biogenic source large enough to accommodate the required changes is agriculture (Saunois et al., 2016). (iii) Any ENSO-driven reduction in biomass burning after 2007 allows for, or requires, growing fossil fuel emissions. The latter has recently been proposed by Worden et al. (2017), who reconstructed larger biomass burning reductions after 2007 than recorded by GFED, although without assigning the reductions to ENSO or other causes.

### 5.3.7. Role of other methane cycle processes

There is an alternative explanation for the lack of correlation between ENSO and the methane records. ENSO could affect $CH_4$ emissions from tropical wetlands and biomass burning as predicted by Hodson et al. (2011) and van der Werf et al. (2006), respectively, but the resulting isotopic signal is overwhelmed by other components of the $CH_4$ cycle. Such influences could be other sources (particularly anthropogenic ones), variability in atmospheric transport or changes in $CH_4$ sink processes. A stronger ENSO signal in Southern tropical $[CH_4]$ and $\delta^{13}CH_4$ compared to Southern mid-latitudes and global average would be expected for several of these scenarios. This is because both biomass burning and wetland emissions show strong maxima in the Southern tropics and should be the dominant sources in this latitudinal band (Kirschke et al., 2013). The detrended $[CH_4]$ records from SMO show such a signal, but one that explains only one third of the IAV and doesn't seem to have significant impact on the trends. Further, we don't find higher ENSO forcing of the $\delta^{13}CH_4$ variability even in the core region of its climatic impact. Corbett et al. (2017) show that during La Niña events high surface temperatures over the Western Pacific lead to upward transport over the Indonesian region (a $CH_4$ source area from wetlands and rice paddies) and negative $CH_4$ anomalies in the mid-troposphere (tropical surface air with relatively low $[CH_4]$ replaces air from the Northern Hemisphere with higher $[CH_4]$). This mechanism would dampen the signal of higher La Niña emissions in surface records like SMO and ASC. However, the corresponding El Niño anomalies in mid-tropospheric $CH_4$ over the Central Pacific are smaller. This indicates that Central Pacific surface air, where there are no $CH_4$ sources, is closer in $[CH_4]$ to mid-tropospheric levels than surface air from the Western Pacific. Unless there were strong longitudinal differences in mid-tropospheric $[CH_4]$, this is inconsistent with a scenario where high concentrations of $CH_4$ are generated over the Western Pacific in La Niñas but transported upwards and away from the surface stations used in this study. On hemispheric or global scales, transport processes are unlikely to play a strong role, given the short mixing time of methane relative to its atmospheric turn-over.

The low correlations of $[CH_4]$ and $\delta^{13}CH_4$ with ENSO rule out a dominant role for ENSO triggered sink changes in atmospheric methane records. Removal processes could lead to either amplification or dampening of source signals. Higher emissions of methane and carbon monoxide from biomass burning will draw down OH and weaken the sink. Emission factors from fires for CO are between 10 and 30-fold higher than for $CH_4$ (Van der Werf et al., 2017), so that the biomass burning dynamics dominate the source of reactive carbon, leaving less OH during El Niños and more during La Niñas to draw down $CH_4$. This would provide a negative feedback for the emissions $[CH_4]$-signal from ENSO forcing. In contrast, the feedback on the ENSO emissions $\delta^{13}CH_4$-signal would be positive due to varying enrichment of $^{13}C$-methane through sink fractionation (less removal leads to less $^{13}C$-enrichment of relatively $^{13}C$-depleted wetland emissions during La Niñas; more removal increases the $^{13}C$-enrichment from biomass burning emissions during El Niños further). In addition to the reactive carbon effect, Turner et al. (2018) found a further OH increase during La Niñas due to higher lightning rates with $NO_x$ production. Turner et al. (2018) could attribute 17% of OH variability that is forced by climate cycles (rather than emissions of other atmospheric compounds) to ENSO. This is a minor part of the variability, but in consequence the dampening effect on $[CH_4]$ and the reinforcing feedback

on $\delta^{13}CH_4$ would add further to the reactive carbon feedbacks. In our correlation results these sink impacts are not apparent, as the $[CH_4]$ correlations for the tropical stations are higher than $\delta^{13}CH_4$ correlations (Tables 2 and 3). Nevertheless, the OH-dynamics provide a possible explanation for the limited ENSO impact on $[CH_4]$ variability and trends. They also make $\delta^{13}CH_4$ a conservative proxy for the influence that ENSO exerts on tropical methane.

Whether ENSO has less influence on $CH_4$ emissions than assumed or whether such an impact is overwhelmed by atmospheric removal or other $CH_4$ cycle processes, our results suggest that global atmospheric trends in $[CH_4]$ and $\delta^{13}CH_4$ are dominated by other components in the methane budget.

## 6. Conclusions

To study the impact of natural climate variability on recent trends in atmospheric methane concentration, we investigated the
correlation between ENSO cycles and records of the mole fractions and stable carbon isotopes of methane, as well as HCN as a biomass burning indicator. As $\delta^{13}CH_4$ is subject to a mutually reinforcing signal from ENSO suppression of wetland emissions and enhancement of biomass burning $CH_4$ (or vice versa), as well as positive feedbacks from OH-dynamics, it is particularly suited to study the role of ENSO in the $CH_4$ cycle.

We find a sizeable effect of ENSO on biomass burning, as indicated by HCN variability in Southern mid-latitudes. In contrast,
ENSO explains a smaller fraction ($\leq 37\%$) of $^[CH_4]$ IAV even in the Southern tropics, where the expected effect should be greatest. Trends in $[CH_4]$ and $\delta^{13}CH_4$ in these latitudes are far less influenced by ENSO ($\leq 23\%$). On hemispheric and global scales the ENSO signal in the methane records is similarly weak. Our results do not rule out that ENSO influences $CH_4$ emissions from wetlands and biomass burning through temperature, enhanced precipitation or droughts in key regions, but any such impacts are overwhelmed by OH-dynamics or other source and sink processes. We review literature estimates of ENSO-
driven emissions and find them too small and sporadic to account for the post-2007 rise. Counteracting OH-dynamics are expected to further dampen any influence ENSO may have on methane growth rates. Our findings suggest that ENSO is not an important driver for recent global trends in methane, including the $[CH_4]$ plateau and the increase in $[CH_4]$ since 2007. The latter must therefore have different causes. Our results do not rule out that wetland production is a contributor to the post-2007 $[CH_4]$-rise if driven by environmental controls other than ENSO. This is suggested by an increase in wetland $CH_4$ production
between the periods 2000-2006 and 2006-2014, although with the limited confidence of a single wetland emissions model (Zhang et al., 2018). The longer the atmospheric $[CH_4]$ and $\delta^{13}CH_4$ trends persist, the less probable are processes that impact IAV and short-lived cyclical events like ENSO as the driver. Therefore, we consider increasing anthropogenic sources as the more likely cause of the $[CH_4]$-rise. Changes in removal rates via OH have been suggested as an additional (Rigby et al., 2017) or alternative (Turner et al., 2017) driver of the increase, but recent work suggests that sink impacts are not dominant (Naus et
al., 2018). There is evidence for additional methane emissions from agriculture (Wolf et al., 2017) and from fossil fuel sources (Hausmann et al., 2016); both may contribute to the current rise in $[CH_4]$ (Worden et al., 2017). Further identification of these processes is necessary to inform climate change mitigation policies and climate projections.

7. Team list

Hinrich Schaefer[1], Dan Smale[1], Sylvia E. Nichol[1], Tony M. Bromley[1], Gordon W. Brailsford[1], Ross J. Martin[1], Rowena Moss[1], Sylvia Englund Michel[2], James W. C. White[2]

[1]Climate and Atmosphere Centre, National Institute of Water and Atmospheric Research, Wellington, 6021, New Zealand
[2] Institute of Arctic and Alpine Research, Boulder, CO, USA.

8. Copyright statement

Code availability: Not applicable

Data availability

Raw data for individual stations measured by INSTAAR and NIWA are available from the World Data Centre for Greenhouse Gases http://ds.data.jma.go.jp/gmd/wdcgg/introduction.html; [$CH_4$] and $\delta^{13}CH_4$ data from NOAA_ESRL and INSTAAR are also available from ftp://aftp.cmdl.noaa.gov/data/trace_gases. Data from Heidelberg University are available from http://www.iup.uni-heidelberg.de/institut/forschung/groups/kk/en/Data_html.
Lauder MIR-FTIR HCN is publicly available from the NDACC archive:
ftp.cpc.ncep.noaa.gov/ndacc/station/lauder/hdf/ftir
All data products and time series used in this study are available on request from the corresponding author.

Appendices

Supplement link (will be included by Copernicus)

Author contribution: H.S. designed the study, conducted the correlation analysis and prepared the manuscript. T.M.B., R.J.M., R.M., G.W.B., S.E.M., and J.W.C.W. supplied $\delta^{13}CH_4$ data. D.S. supplied HCN data. S.E.N. conducted data processing and analysis.

**14. Acknowledgements**

We thank B. Graham, E. Behrens, Brett Mullan and S. Mikaloff Fletcher for helpful discussions and advice. Ed Dlugokencky from NOAA ESRL supplied [$CH_4$] data. D. Lowe started the BHD measurements that made this analysis possible. This project was supported by NIWA funding under Climate and Atmosphere Research Programme CAAC1804 (2017/18 SCI).

**15. Disclaimer**

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

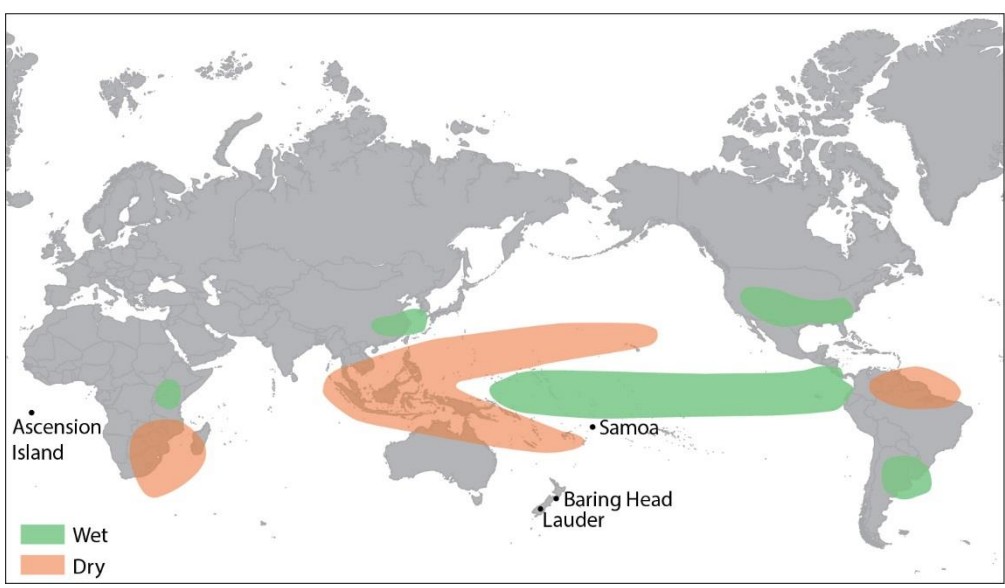

Fig. 1: Regions of ENSO impacts and monitoring stations used in this study.

5  The map indicates the locations of the atmospheric monitoring stations on Ascension Island (ASC), Samoa (SMO), Baring Head (BHD) and Lauder (LAU). General precipitation anomalies during northern hemisphere El Niño conditions for Dec-Feb are taken from https://www.climate.gov/news-features/featured-images/global-impacts-el-nino-and-la-nina. El Niño dry regions in Jun-Aug are similar for southern Asia and South America; during La Niña events opposite patterns for wet- and dryness develop in roughly the same regions.

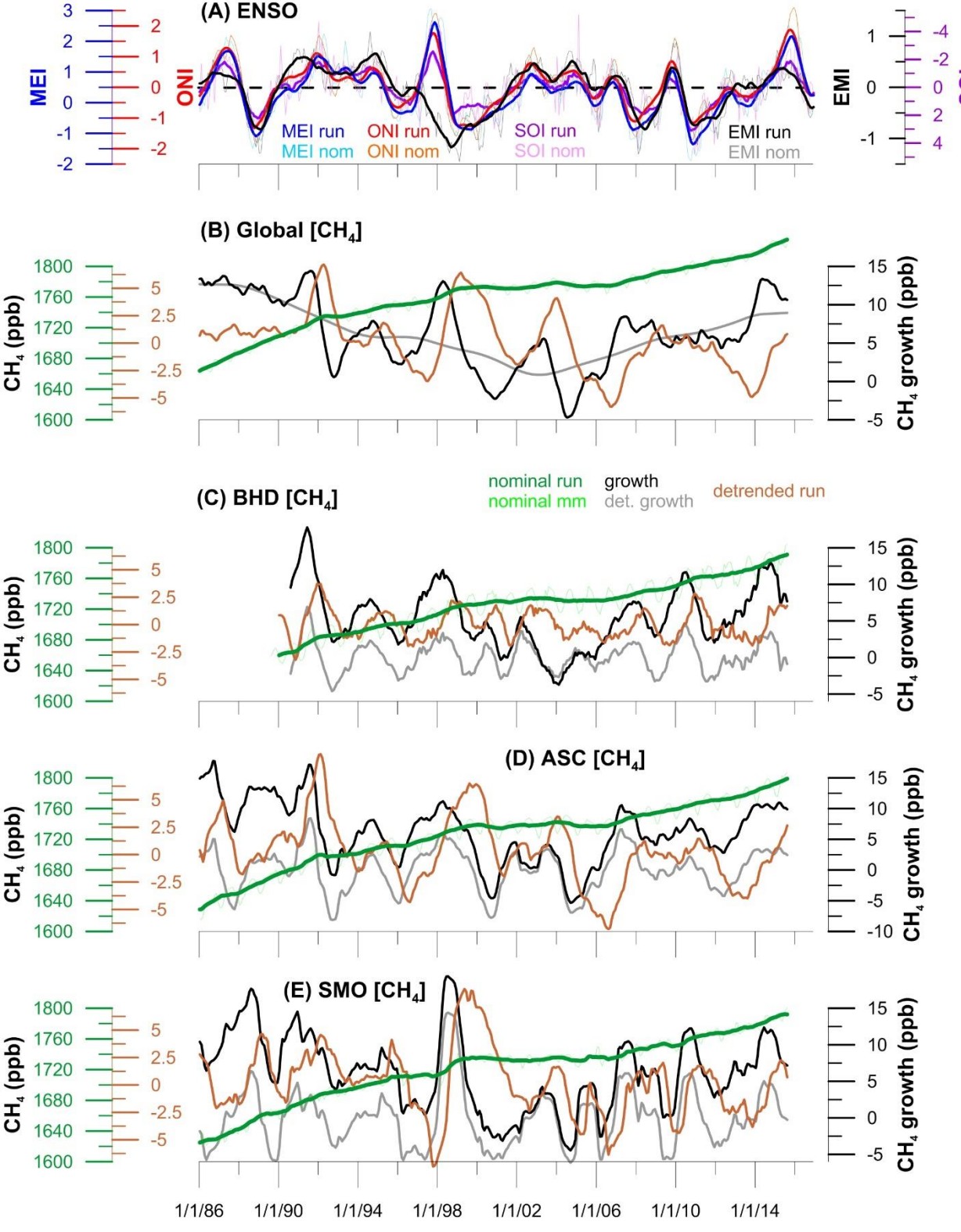

Fig. 2: Selected time series of ENSO indices and [CH$_4$]

Panels from top to bottom: (A) multivariate ENSO Index (MEI), Southern Oscillation Index (SOI), Ocean Niño Index (ONI), and El Modoki Index (EMI) shown for nominal literature data and their 12-month running means. (B) Global [CH$_4$] records; monthly means, 12-month running mean, detrended 12-month running mean, as well as nominal and detrended growth rates. (C) [CH$_4$] records from BHD (D) [CH$_4$] records from ASC. (E) [CH$_4$] records from SMO. BHD, ASC and SMO display same records as for global time series.

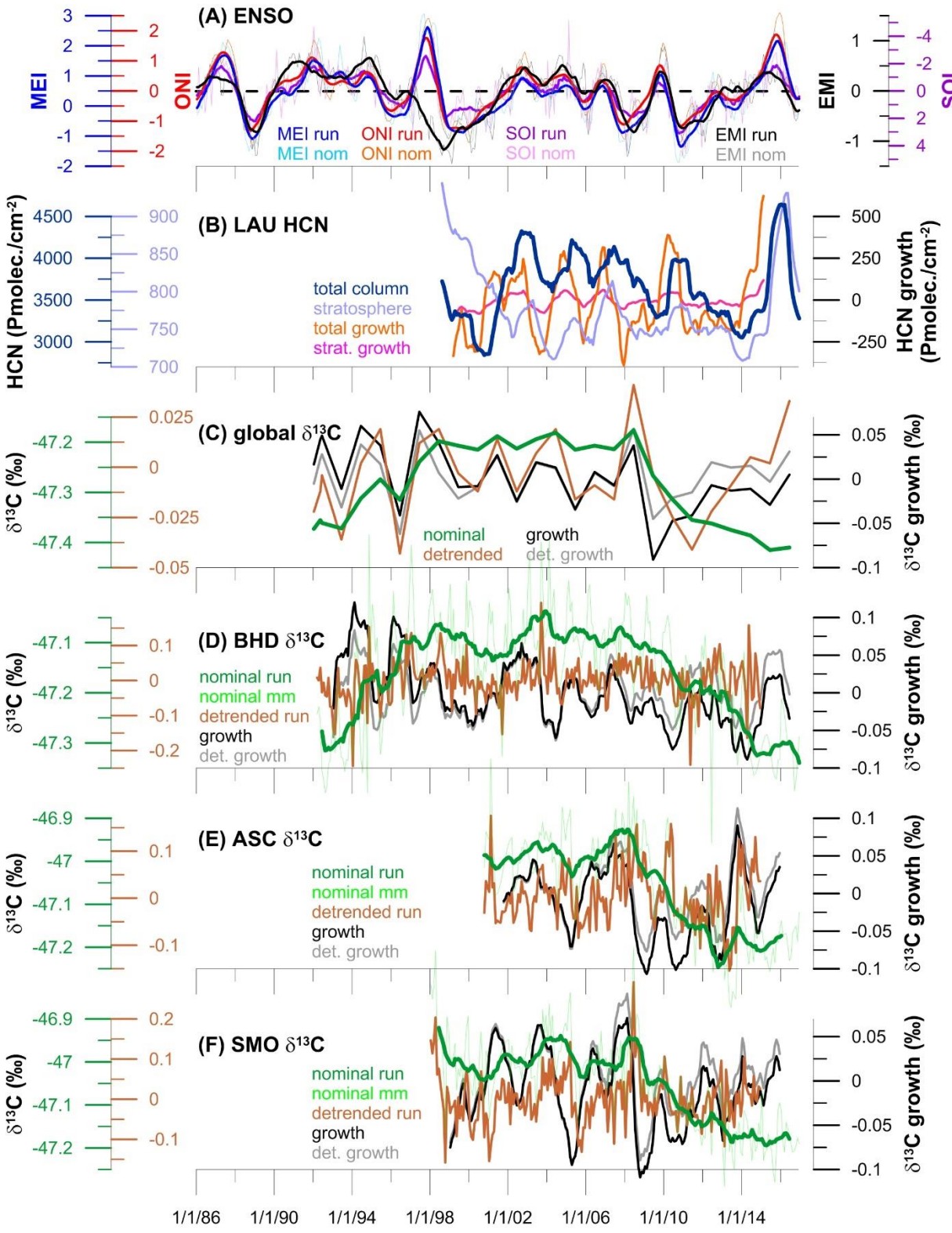

Fig. 3: Selected time series of ENSO indices, HCN and $\delta^{13}CH_4$

Panels from top to bottom: (A) multivariate ENSO Index (MEI), Southern Oscillation Index (SOI), Ocean Niño Index (ONI), and El Modoki Index (EMI) shown for nominal literature data and their 12-month running means. (B) HCN records as 12-month running means from LAU for total atmospheric column and stratosphere (12-100 km) and respective growth rates. (C) Global annually averaged $\delta^{13}CH_4$ according to Schaefer et al. (2016) updated to end of 2016; nominal and detrended values and their respective growth rates. (D) $\delta^{13}CH_4$ from BHD; monthly means, 12-month running mean, detrended 12-month running mean, as well as nominal and detrended growth rates. (E) $\delta^{13}CH_4$ from ASC. (F) $\delta^{13}CH_4$ from SMO. ASC and SMO display same records as for BHD. Note scale differences between all $\delta^{13}C$-axes to accentuate variability for comparison with ENSO.

**Table 1: Description of time series products used in the correlation analyses.**

| Parameter | Time series | Description |
|---|---|---|
| [CH$_4$] | global nom | global monthly means |
| | global gro | global monthly growth rates |
| | global run | 12-month running mean of global monthly means |
| | global res | STL residual of global monthly means |
| | glob det-nom | detrended global monthly means |
| | glob det-gro | detrended global monthly growth rates |
| | glob det-run | detrended 12-month running mean of global monthly means |
| $\delta^{13}CH_4$ and [CH$_4$] | NNN* nom | station monthly means |
| | NNN* gro | station monthly growth rates |
| | NNN* run | 12-month running mean of station monthly means |
| | NNN* res | STL residual of station monthly means |
| | NNN* det-nom | detrended station monthly means |
| | NNN* det-gro | detrended station monthly growth rates |
| | NNN* det-run | detrended 12-month running mean of station monthly means |
| HCN (LAU) | total nom | Total column monthly means |
| | total gro | Total column monthly growth rates |
| | total run | Total column 12-month running mean of station monthly means |
| | total res | Total column STL residual of station monthly means |
| | strat. nom | Stratosphere monthly means |
| | strat. gro | Stratosphere monthly growth rates |
| | strat. run | Stratosphere 12-month running mean of station monthly means |
| | strat. res | Stratosphere STL residual of station monthly means |
| $\delta^{13}CH_4$ | global nom | global yearly means |
| | global gro | global yearly growth rates |
| | global det | detrended global yearly means |
| | global det-gro | detrended global yearly growth rates |

*NNN as station acronym ASC, BHD, or SMO

**Table 2: Spearman correlation of methane mole fraction with ENSO variability.**

Correlations (r²-values) for the Spearman ranking coefficient between [CH₄] time series from various sites and ENSO indices with lag times (in months) for optimum results. Colour backgrounds indicate r²-values in 10% classes. Grey background indicates correlations with p-values > 0.001.

| Time series | MEI nom r² | lag | MEI run r² | lag | ONI nom r² | lag | ONI run r² | lag | SOI nom r² | lag | SOI run r² | lag | EMI nom r² | lag | EMI run r² | lag |
|---|---|---|---|---|---|---|---|---|---|---|---|---|---|---|---|---|
| **[CH₄]** | | | | | | | | | | | | | | | | |
| global nom | 0.08 | 59 | 0.12 | 58 | 0.04 | 48 | 0.06 | 50 | 0.10 | 59 | 0.17 | 54 | 0.04 | 5 | 0.03 | 10 |
| global gro | 0.10 | 6 | 0.10 | 6 | 0.08 | 7 | 0.09 | 59 | 0.06 | 8 | 0.08 | 33 | 0.06 | 50 | 0.09 | 52 |
| global run | 0.10 | 56 | 0.18 | 53 | 0.06 | 55 | 0.09 | 51 | 0.11 | 57 | 0.24 | 54 | 0.07 | 8 | 0.08 | 10 |
| global res | 0.06 | 49 | 0.06 | 47 | 0.09 | 49 | 0.11 | 48 | 0.04 | 25 | 0.06 | 60 | 0.10 | 0 | 0.09 | 0 |
| glob det-nom | 0.10 | 49 | 0.02 | 49 | 0.06 | 49 | 0.04 | 50 | 0.05 | 29 | 0.02 | 60 | 0.07 | 9 | 0.03 | 5 |
| glob det-gro | 0.02 | 58 | 0.04 | 60 | 0.00 | 58 | 0.00 | 59 | 0.02 | 59 | 0.06 | 60 | 0.03 | 0 | 0.04 | 0 |
| glob det-run | 0.05 | 49 | 0.06 | 48 | 0.09 | 48 | 0.12 | 49 | 0.03 | 0 | 0.04 | 60 | 0.08 | 0 | 0.08 | 1 |
| BHD nom | 0.10 | 56 | 0.11 | 58 | 0.05 | 45 | 0.05 | 47 | 0.11 | 56 | 0.15 | 59 | 0.07 | 53 | 0.05 | 51 |
| BHD gro | 0.08 | 24 | 0.10 | 27 | 0.10 | 51 | 0.11 | 55 | 0.08 | 24 | 0.12 | 24 | 0.15 | 60 | 0.19 | 60 |
| BHD run | 0.05 | 44 | 0.11 | 44 | 0.04 | 0 | 0.07 | 0 | 0.08 | 45 | 0.17 | 53 | 0.08 | 7 | 0.09 | 7 |
| BHD res | 0.03 | 46 | 0.02 | 16 | 0.02 | 16 | 0.02 | 16 | 0.02 | 16 | 0.02 | 14 | 0.02 | 33 | 0.02 | 29 |
| BHD det-nom | 0.07 | 44 | 0.01 | 17 | 0.02 | 33 | 0.01 | 36 | 0.03 | 13 | 0.01 | 17 | 0.04 | 30.00 | 0.00 | 29 |
| BHD det-gro | 0.13 | 56 | 0.17 | 58 | 0.12 | 23 | 0.13 | 59 | 0.10 | 23 | 0.14 | 59 | 0.08 | 60 | 0.08 | 24 |
| BHD det-run | 0.10 | 60 | 0.14 | 60 | 0.07 | 44 | 0.09 | 42 | 0.07 | 60 | 0.11 | 60 | 0.08 | 33 | 0.11 | 32 |
| ASC nom | 0.09 | 56 | 0.11 | 45 | 0.05 | 44 | 0.06 | 46 | 0.10 | 42 | 0.16 | 46 | 0.05 | 53 | 0.04 | 50 |
| ASC gro | 0.09 | 29 | 0.13 | 31 | 0.11 | 53 | 0.13 | 55 | 0.06 | 31 | 0.11 | 32 | 0.15 | 50 | 0.21 | 53 |
| ASC run | 0.08 | 43 | 0.16 | 45 | 0.07 | 44 | 0.10 | 44 | 0.11 | 43 | 0.22 | 43 | 0.06 | 46 | 0.06 | 47 |
| ASC res | 0.08 | 42 | 0.09 | 18 | 0.12 | 42 | 0.12 | 42 | 0.06 | 41 | 0.06 | 60 | 0.08 | 0 | 0.07 | 2 |
| ASC det-nom | 0.11 | 43 | 0.02 | 45 | 0.06 | 43 | 0.03 | 45 | 0.06 | 43 | 0.01 | 44 | 0.06 | 4 | 0.02 | 2 |
| ASC det-gro | 0.20 | 10 | 0.26 | 10 | 0.18 | 10 | 0.21 | 11 | 0.14 | 10 | 0.20 | 10 | 0.09 | 51 | 0.12 | 53 |
| ASC det-run | 0.09 | 40 | 0.14 | 17 | 0.13 | 41 | 0.15 | 42 | 0.05 | 17 | 0.08 | 17 | 0.09 | 0 | 0.08 | 1 |
| SMO nom | 0.07 | 56 | 0.12 | 58 | 0.03 | 45 | 0.04 | 48 | 0.08 | 56 | 0.16 | 59 | 0.02 | 10 | 0.02 | 50 |
| SMO gro | 0.19 | 17 | 0.18 | 10 | 0.15 | 10 | 0.17 | 11 | 0.10 | 9 | 0.12 | 10 | 0.13 | 46 | 0.17 | 49 |
| SMO run | 0.07 | 51 | 0.14 | 53 | 0.04 | 49 | 0.05 | 51 | 0.08 | 53 | 0.18 | 55 | 0.01 | 7 | 0.01 | 10 |
| SMO res | 0.16 | 0 | 0.13 | 1 | 0.22 | 1 | 0.15 | 1 | 0.14 | 0 | 0.15 | 2 | 0.17 | 0 | 0.14 | 3 |
| SMO det-nom | 0.15 | 0 | 0.11 | 1 | 0.18 | 0 | 0.14 | 1 | 0.09 | 0 | 0.12 | 2 | 0.10 | 0 | 0.11 | 3 |
| SMO det-gro | 0.31 | 9 | 0.36 | 10 | 0.31 | 10 | 0.35 | 11 | 0.22 | 9 | 0.33 | 11 | 0.10 | 10 | 0.11 | 12 |
| SMO det-run | 0.26 | 1 | 0.24 | 2 | 0.29 | 2 | 0.27 | 3 | 0.21 | 2 | 0.25 | 2 | 0.23 | 2 | 0.21 | 4 |

**Table 3: Spearman correlation of $\delta^{13}CH_4$ and HCN with ENSO variability.**

5   Correlations ($r^2$-values) for the Spearman ranking coefficient between dependent variables, i.e. $\delta^{13}CH_4$ and HCN time series from various sites, and ENSO indices with lag times (in months) for optimum results. Colour backgrounds indicate $r^2$-values in 10% classes. Grey background indicates correlations with p-values > 0.001.

| Time series | MEI nom $r^2$ | lag | MEI run $r^2$ | lag | ONI nom $r^2$ | lag | ONI run $r^2$ | lag | SOI nom $r^2$ | lag | SOI run $r^2$ | lag | EMI nom $r^2$ | lag | EMI run $r^2$ | lag |
|---|---|---|---|---|---|---|---|---|---|---|---|---|---|---|---|---|
| **HCN (LAU)** | | | | | | | | | | | | | | | | |
| total nom | 0.13 | 5 | 0.03 | 6 | 0.06 | 5 | 0.05 | 8 | 0.08 | 4 | 0.04 | 8 | 0.10 | 14 | 0.05 | 12 |
| total gro | 0.21 | 1 | 0.26 | 2 | 0.27 | 1 | 0.27 | 2 | 0.18 | 0 | 0.30 | 1 | 0.19 | 0 | 0.27 | 1 |
| total run | 0.23 | 7 | 0.30 | 9 | 0.35 | 7 | 0.39 | 9 | 0.23 | 8 | 0.38 | 8 | 0.40 | 9 | 0.51 | 10 |
| total res | 0.10 | 7 | 0.09 | 7 | 0.13 | 7 | 0.13 | 7 | 0.10 | 10 | 0.13 | 6 | 0.08 | 9 | 0.10 | 10 |
| strat. nom | 0.10 | 10 | 0.05 | 11 | 0.05 | 41 | 0.03 | 0 | 0.08 | 40 | 0.05 | 43 | 0.09 | 39 | 0.08 | 0 |
| strat. gro | 0.36 | 0 | 0.43 | 0 | 0.40 | 0 | 0.44 | 1 | 0.22 | 0 | 0.37 | 0 | 0.21 | 0 | 0.30 | 0 |
| strat. run | 0.03 | 37 | 0.05 | 38 | 0.05 | 0 | 0.08 | 0 | 0.05 | 40 | 0.08 | 41 | 0.07 | 18 | 0.10 | 0 |
| strat. res | 0.10 | 7 | 0.07 | 7 | 0.10 | 7 | 0.08 | 8 | 0.07 | 8 | 0.07 | 8 | 0.11 | 44 | 0.10 | 38 |
| **$\delta^{13}CH_4$** | | | | | | | | | | | | | | | | |
| global nom | 0.17 | 2 | 0.17 | 49 | 0.13 | 2 | 0.12 | 50 | 0.27 | 46 | 0.27 | 49 | 0.16 | 4 | 0.05 | 47 |
| global gro | 0.20 | 39 | 0.18 | 41 | 0.18 | 39 | 0.16 | 41 | 0.34 | 49 | 0.20 | 15 | 0.15 | 4 | 0.08 | 16 |
| global det | 0.14 | 1 | 0.19 | 14 | 0.08 | 20 | 0.12 | 16 | 0.37 | 22 | 0.27 | 17 | 0.13 | 15 | 0.12 | 16 |
| global det-gro | 0.23 | 58 | 0.16 | 58 | 0.18 | 59 | 0.18 | 60 | 0.37 | 55 | 0.24 | 56 | 0.15 | 15 | 0.12 | 16 |
| BHD nom | 0.09 | 56 | 0.09 | 60 | 0.04 | 56 | 0.04 | 59 | 0.09 | 55 | 0.13 | 60 | 0.07 | 42 | 0.06 | 39 |
| BHD gro | 0.05 | 2 | 0.06 | 2 | 0.06 | 2 | 0.07 | 3 | 0.08 | 2 | 0.13 | 2 | 0.10 | 44 | 0.11 | 44 |
| BHD run | 0.09 | 14 | 0.14 | 16 | 0.11 | 15 | 0.15 | 17 | 0.10 | 15 | 0.20 | 17 | 0.07 | 29 | 0.09 | 36 |
| BHD res | 0.03 | 6 | 0.03 | 10 | 0.04 | 7 | 0.03 | 9 | 0.04 | 6 | 0.03 | 10 | 0.09 | 33 | 0.09 | 36 |
| BHD det-nom | 0.07 | 8 | 0.01 | 8 | 0.03 | 7 | 0.02 | 9 | 0.05 | 6 | 0.01 | 10 | 0.06 | 42 | 0.05 | 38 |
| BHD det-gro | 0.07 | 1 | 0.09 | 1 | 0.09 | 1 | 0.11 | 2 | 0.09 | 2 | 0.13 | 12 | 0.10 | 45 | 0.11 | 45 |
| BHD det-run | 0.07 | 9 | 0.09 | 11 | 0.08 | 12 | 0.10 | 12 | 0.06 | 9 | 0.09 | 13 | 0.15 | 32 | 0.20 | 35 |
| ASC nom | 0.08 | 54 | 0.08 | 44 | 0.05 | 54 | 0.04 | 45 | 0.08 | 55 | 0.13 | 45 | 0.05 | 52 | 0.03 | 49 |
| ASC gro | 0.10 | 13 | 0.14 | 13 | 0.10 | 13 | 0.09 | 13 | 0.11 | 12 | 0.22 | 12 | 0.13 | 9 | 0.13 | 7 |
| ASC run | 0.08 | 22 | 0.13 | 23 | 0.08 | 22 | 0.10 | 22 | 0.12 | 22 | 0.23 | 22 | 0.12 | 20 | 0.14 | 20 |
| ASC res | 0.03 | 17 | 0.04 | 21 | 0.03 | 16 | 0.02 | 19 | 0.05 | 18 | 0.06 | 20 | 0.03 | 14 | 0.03 | 19 |
| ASC det-nom | 0.05 | 18 | 0.03 | 22 | 0.03 | 17 | 0.01 | 19 | 0.06 | 18 | 0.04 | 17 | 0.05 | 14 | 0.01 | 16 |

| | | | | | | | | | | | | | | | | |
|---|---|---|---|---|---|---|---|---|---|---|---|---|---|---|---|---|
| ASC det-gro | 0.07 | 13 | 0.07 | 14 | 0.05 | 13 | 0.05 | 60 | 0.06 | 12 | 0.11 | 13 | 0.07 | 8 | 0.05 | 6 |
| ASC det-run | 0.04 | 22 | 0.05 | 60 | 0.04 | 60 | 0.04 | 60 | 0.07 | 22 | 0.09 | 22 | 0.06 | 32 | 0.06 | 32 |
| SMO nom | 0.06 | 56 | 0.08 | 55 | 0.04 | 43 | 0.04 | 44 | 0.08 | 56 | 0.12 | 42 | 0.04 | 40 | 0.05 | 38 |
| SMO gro | 0.02 | 13 | 0.03 | 24 | 0.02 | 23 | 0.03 | 25 | 0.04 | 0 | 0.06 | 13 | 0.10 | 43 | 0.10 | 44 |
| SMO run | 0.07 | 15 | 0.12 | 18 | 0.08 | 16 | 0.10 | 19 | 0.09 | 16 | 0.19 | 20 | 0.06 | 19 | 0.07 | 21 |
| SMO res | 0.09 | 0 | 0.08 | 0 | 0.10 | 1 | 0.08 | 2 | 0.09 | 1 | 0.07 | 3 | 0.06 | 30 | 0.06 | 35 |
| SMO det-nom | 0.06 | 0 | 0.06 | 1 | 0.08 | 1 | 0.06 | 1 | 0.06 | 3 | 0.05 | 3 | 0.06 | 30 | 0.05 | 35 |
| SMO det-gro | 0.05 | 23 | 0.06 | 24 | 0.04 | 23 | 0.06 | 25 | 0.03 | 0 | 0.03 | 0 | 0.09 | 42 | 0.08 | 42 |
| SMO det-run | 0.17 | 1 | 0.20 | 1 | 0.20 | 2 | 0.22 | 2 | 0.12 | 4 | 0.16 | 3 | 0.13 | 26 | 0.14 | 30 |

**Table 4: Pearson correlation of $\delta^{13}CH_4$ and HCN with ENSO variability.**

Correlations ($r^2$-values) for the Pearson correlation coefficient between dependent variables, i.e. $\delta^{13}CH_4$ and HCN time series from various sites, and ENSO indices with lag times (in months) for optimum results. Colour backgrounds indicate $r^2$-values in 10% classes. Results have not been screened for p-values.

| Time series | MEI nom | | MEI run | | ONI nom | | ONI run | | SOI nom | | SOI run | | EMI nom | | EMI run | |
|---|---|---|---|---|---|---|---|---|---|---|---|---|---|---|---|---|
| | $r^2$ | lag | $r^2$ | lag | $r^2$ | lag | $r^2$ | lag | $r^2$ | lag | $r^2$ | lag | $r^2$ | lag | $r^2$ | lag |
| **HCN (LAU)** | | | | | | | | | | | | | | | | |
| total nom | 0.10 | 5 | 0.03 | 5 | 0.04 | 4 | 0.04 | 6 | 0.06 | 4 | 0.04 | 6 | 0.10 | 14 | 0.06 | 11 |
| total gro | 0.22 | 0 | 0.30 | 2 | 0.27 | 1 | 0.30 | 1 | 0.16 | 0 | 0.28 | 2 | 0.19 | 2 | 0.24 | 1 |
| total run | 0.22 | 4 | 0.29 | 6 | 0.34 | 6 | 0.40 | 7 | 0.18 | 6 | 0.34 | 8 | 0.36 | 10 | 0.46 | 11 |
| total res | 0.11 | 7 | 0.13 | 5 | 0.14 | 5 | 0.16 | 6 | 0.09 | 7 | 0.13 | 7 | 0.09 | 10 | 0.08 | 11 |
| strat. nom | 0.18 | 6 | 0.13 | 9 | 0.08 | 7 | 0.08 | 9 | 0.07 | 5 | 0.09 | 9 | 0.09 | 39 | 0.11 | 0 |
| strat. gro | 0.42 | 0 | 0.54 | 0 | 0.49 | 0 | 0.55 | 0 | 0.22 | 0 | 0.39 | 0 | 0.18 | 1 | 0.25 | 1 |
| strat. run | 0.12 | 14 | 0.17 | 12 | 0.05 | 11 | 0.07 | 13 | 0.05 | 17 | 0.11 | 13 | 0.18 | 0 | 0.18 | 0 |
| strat. res | 0.17 | 6 | 0.16 | 6 | 0.17 | 6 | 0.17 | 7 | 0.08 | 5 | 0.12 | 8 | 0.10 | 39 | 0.09 | 37 |
| **$\delta^{13}CH_4$** | | | | | | | | | | | | | | | | |
| global nom | 0.16 | 1 | 0.16 | 50 | 0.14 | 2 | 0.09 | 50 | 0.24 | 46 | 0.25 | 39 | 0.13 | 3 | 0.05 | 50 |
| global gro | 0.20 | 39 | 0.14 | 42 | 0.16 | 39 | 0.13 | 43 | 0.28 | 49 | 0.20 | 15 | 0.12 | 41 | 0.10 | 0 |
| global det | 0.18 | 11 | 0.15 | 13 | 0.15 | 11 | 0.12 | 13 | 0.32 | 12 | 0.24 | 15 | 0.11 | 15 | 0.08 | 17 |
| glob det-gro | 0.14 | 40 | 0.13 | 58 | 0.13 | 58 | 0.12 | 59 | 0.28 | 49 | 0.14 | 59 | 0.16 | 15 | 0.10 | 16 |
| BHD nom | 0.10 | 56 | 0.09 | 60 | 0.04 | 0 | 0.05 | 0 | 0.10 | 55 | 0.14 | 60 | 0.05 | 42 | 0.04 | 39 |
| BHD gro | 0.07 | 61 | 0.10 | 60 | 0.08 | 0 | 0.09 | 1 | 0.08 | 2 | 0.14 | 1 | 0.14 | 45 | 0.17 | 47 |
| BHD run | 0.10 | 0 | 0.13 | 0 | 0.11 | 0 | 0.13 | 0 | 0.09 | 62 | 0.18 | 57 | 0.04 | 38 | 0.06 | 41 |
| BHD res | 0.05 | 6 | 0.05 | 9 | 0.06 | 7 | 0.05 | 9 | 0.06 | 6 | 0.04 | 10 | 0.07 | 35 | 0.05 | 9 |
| BHD det-nom | 0.07 | 8 | 0.02 | 7 | 0.03 | 7 | 0.02 | 8 | 0.06 | 6 | 0.02 | 9 | 0.05 | 42 | 0.04 | 34 |
| BHD det-gro | 0.10 | 0 | 0.13 | 0 | 0.13 | 0 | 0.15 | 1 | 0.07 | 2 | 0.13 | 0 | 0.12 | 45 | 0.14 | 47 |
| BHD det-run | 0.10 | 9 | 0.13 | 10 | 0.11 | 10 | 0.14 | 12 | 0.06 | 9 | 0.10 | 12 | 0.18 | 33 | 0.23 | 34 |
| ASC nom | 0.09 | 54 | 0.08 | 58 | 0.04 | 54 | 0.04 | 37 | 0.09 | 55 | 0.16 | 33 | 0.04 | 52 | 0.03 | 32 |
| ASC gro | 0.10 | 14 | 0.14 | 14 | 0.08 | 13 | 0.10 | 13 | 0.08 | 12 | 0.19 | 13 | 0.15 | 5 | 0.22 | 7 |
| ASC run | 0.07 | 51 | 0.11 | 54 | 0.04 | 50 | 0.06 | 53 | 0.09 | 51 | 0.20 | 27 | 0.05 | 22 | 0.08 | 23 |
| ASC res | 0.03 | 17 | 0.04 | 22 | 0.03 | 18 | 0.02 | 19 | 0.05 | 29 | 0.07 | 24 | 0.04 | 14 | 0.03 | 18 |
| ASC det-nom | 0.06 | 18 | 0.04 | 22 | 0.03 | 18 | 0.01 | 18 | 0.06 | 29 | 0.05 | 32 | 0.05 | 15 | 0.02 | 17 |
| ASC det-gro | 0.07 | 14 | 0.08 | 14 | 0.04 | 13 | 0.05 | 59 | 0.05 | 13 | 0.09 | 14 | 0.07 | 8 | 0.10 | 7 |
| ASC det-run | 0.07 | 22 | 0.09 | 23 | 0.04 | 59 | 0.05 | 59 | 0.07 | 25 | 0.16 | 24 | 0.06 | 56 | 0.08 | 57 |
| SMO nom | 0.07 | 56 | 0.08 | 54 | 0.03 | 45 | 0.04 | 45 | 0.09 | 42 | 0.14 | 45 | 0.05 | 41 | 0.05 | 39 |
| SMO gro | 0.04 | 66 | 0.04 | 67 | 0.04 | 66 | 0.03 | 25 | 0.03 | 0 | 0.06 | 0 | 0.10 | 43 | 0.11 | 43 |
| SMO run | 0.07 | 50 | 0.10 | 49 | 0.03 | 38 | 0.05 | 0 | 0.09 | 51 | 0.18 | 49 | 0.03 | 38 | 0.04 | 35 |
| SMO res | 0.10 | 0 | 0.07 | 2 | 0.10 | 1 | 0.07 | 3 | 0.10 | 1 | 0.06 | 5 | 0.06 | 14 | 0.06 | 16 |
| SMO det-nom | 0.07 | 0 | 0.06 | 2 | 0.08 | 1 | 0.05 | 3 | 0.07 | 1 | 0.05 | 5 | 0.06 | 14 | 0.05 | 16 |
| SMO det-gro | 0.05 | 22 | 0.05 | 24 | 0.05 | 66 | 0.06 | 25 | 0.02 | 26 | 0.03 | 26 | 0.06 | 42 | 0.07 | 42 |
| SMO det-run | 0.06 | 1 | 0.06 | 2 | 0.07 | 2 | 0.08 | 3 | 0.05 | 15 | 0.07 | 16 | 0.14 | 16 | 0.19 | 20 |