# Peer review of "Limited impact of El Niño – Southern Oscillation on variability and growth rate of atmospheric methane"

_Biogeosciences, 2018_

## Referee Comment (RC1) · Anonymous Referee #1 · 7 Aug 2018

The paper by Schaefer et al investigates the role of ENSO anomalies on atmospheric methane concentrations, with additional insights provided by investigating its isotopologue (13C-CH4) and hydrogen cyanide as a proxy for fire. Using a diverse set of ENSO indicators, with a variety of approaches for smoothing and integrating temporal lags, the authors find that ENSO has a small role on atmospheric CH4 concentrations, and conclude that ENSO has played only a small role on the renewed growth in concentrations since 2006.

The manuscript is an important contribution in terms of the renewed growth discussion of atmospheric methane concentrations because it provides additional evidence that emissions sources that have high interannual-variability, i.e., wetlands and fire, are unlikely to be the dominant cause of sustained emissions. However, while the methods,

results and discussion are fairly clear, the title and the Introduction could be clarified to reflect the main message.

First, I recommend the authors revisit the title and modify to be more specific than just 'methane cycle' because this implies the authors were looking at methane emissions, but rather the authors investigated atmospheric concentrations. I would prefer a title along the lines of "Limited impact of El Niño – Southern Oscillation on the atmospheric methane growth anomalies"

Second, the Introduction could be clearer to reflect that the authors are motivated by understanding atmospheric methane concentration anomalies rather than anomalies in emissions. The previous studies linking methane emissions to ENSO as a key driver are not in question, but currently the Introduction mixes a little the emissions and concentrations anomalies making the reader have to work to clarify this.

In Table 1, I assume the lag time is in months, so 54 is a 54 month lag? If so, many are longer than 12 months, which is contrary to the statement in Section 5.2 that says most are shorter than a year. I am skeptical of such long lags, it is difficult to judge whether shorter lags were close in terms of significance to the longer lag times because these numbers are not presented.

I commend the authors on the discussion of transport and atmospheric mixing, it was very helpful to have this context while thinking about the correlations and locations of sampling stations.

---

## Referee Comment (RC2) · J. R. Melton (Referee) · 28 Aug 2018

Review of Schaefer et al. for BG

Schaefer and coworkers investigate the influence of ENSO on the methane cycle. They use methane concentration records both globally and at several sites along with d13CH4 and HCN (for biomass burning) records. The d13CH4 records are helpful as they can better constrain the methane sources since pyrogenic sources are very 13C-enriched while biogenic methane is usually 13C-depleted with thermogenic sources somewhere in between. Their study uses correlation analysis to attempt to tease out the impact of ENSO on the methane cycle but they find little evident influence. The paper is generally well written and I think that the overall story is of interest to the com-

**BGD**

munity as there have been other papers using other means to suggest that parts of the methane cycle (particularly wetlands and fire) are influenced by ENSO. There are some typos etc. that I leave to a copy editor. I have a few comments below but I think that the paper should be published after some revisions to address some questions/concerns.

Main comments:

1. I didn't see a discussion of significance level for the correlation coefficients. Without one I have trouble understanding if a value of 0.25 is significant or not. Is there a reason that was not done? Otherwise, while all of the r2 are low, it might help understand what are just noise and what is representing a true signal.

2. There is relatively little discussion on possible changes in the main sink of methane - hydroxyl radicals. The dynamics of this sink has been highlighted in recent studies (McNorton et al. 2016, Turner et al. 2017, Rigby et al. 2017). Is it possible that ENSO would have an impact upon local concentrations of OH? Is it a safe assumption to assume a constant sink strength? Given the power of this sink, and its recent importance (at least in the studies just mentioned) perhaps it is worthwhile to give more consideration to how a changing sink due to ENSO could impact upon the methane cycle. Or at least expand the discussion of the sink to demonstrate why a constant assumption is valid.

Smaller comments:

- What about using the GFED4(s) burned area product to investigate changes in burning? It would have the advantage that it is global and extends further back in time than the HCN record. However the caveat is that burned area is not the same as C emitted as CH4, however it might be a reasonable test since they would be closely related. A recent study highlighted the continual decrease in burnt area over the course of the record (Andela et al. 2017) which then should have some impact upon the methane cycle and perhaps can be used to tease out any ENSO influence.

- Can we have a table with the various tests laid out (det-nom, det-gro, nominal mm, nominal run, etc.)? It is difficult to keep them all in the head and then interpret some very busy figures.

- p. 2 line 30 - What is aggregate source here?

- Author contributions - there is no E.D author.

- Data avail - raw data is little use. Best approach would be to make the actual analysis available.

Refs cited:

McNorton, J., Chipperfield, M. P., Gloor, M., Wilson, C., Feng, W., Hayman, G. D., Rigby, M., Krummel, P. B., O'Doherty, S., Prinn, R. G., Weiss, R. F., Young, D., Dlugokencky, E. and Montzka, S. A.: Role of OH variability in the stalling of the global atmospheric CH4 growth rate from 1999 to 2006, , doi:10.5194/acp-16-7943-2016, 2016.

Turner, A. J., Frankenberg, C., Wennberg, P. O. and Jacob, D. J.: Ambiguity in the causes for decadal trends in atmospheric methane and hydroxyl, Proc. Natl. Acad. Sci. U. S. A., doi:10.1073/pnas.1616020114, 2017.

Rigby, M., Montzka, S. A., Prinn, R. G., White, J. W. C., Young, D., O'Doherty, S., Lunt, M. F., Ganesan, A. L., Manning, A. J., Simmonds, P. G., Salameh, P. K., Harth, C. M., Mühle, J., Weiss, R. F., Fraser, P. J., Steele, L. P., Krummel, P. B., McCulloch, A. and Park, S.: Role of atmospheric oxidation in recent methane growth, Proc. Natl. Acad. Sci. U. S. A., doi:10.1073/pnas.1616426114, 2017.

Andela, N., Morton, D. C., Giglio, L., Chen, Y., van der Werf, G. R., Kasibhatla, P. S., DeFries, R. S., Collatz, G. J., Hantson, S., Kloster, S., Bachelet, D., Forrest, M., Lasslop, G., Li, F., Mangeon, S., Melton, J. R., Yue, C. and Randerson, J. T.: A human-driven decline in global burned area, Science, 356(6345), 1356–1362, 2017.

---

## Short Comment (SC1) · 28 Aug 2018

Dear editor,

I was recently pointed to this manuscript on the role of ENSO on the methane cycle. The manuscript argues for a limited role of ENSO on the methane cycle; however, the manuscript makes little mention of two important factors that impact atmospheric methane and are strongly influenced by ENSO: **(1) atmospheric transport** and **(2) loss via hydroxyl**. These factors seem particularly pertinent to a discussion of the role of ENSO on the methane cycle. There have been a number of recent papers on these two topics in the last two years that the authors seem to have overlooked. Specifically, McNorton *et al.* (2016), Turner *et al.* (2017), and Rigby *et al.* (2017) showed

how changes in the methane loss via oxidation by hydroxyl was an important factor in the interpretation of methane trends. More directly related to ENSO, Corbett *et al.* (2017) showed the influence of ENSO on the spatial distribution of methane via changes in atmospheric transport while Turner *et al.* (2018) showed how ENSO can strongly influence the methane lifetime.

- **McNorton *et al.*, ACP (2016):** *"Role of OH variability in the stalling of the global atmospheric $CH_4$ growth rate from 1999 to 2006"*, https://doi.org/10.5194/acp-16-7943-2016.

- **Corbett *et al.*, GRL (2017):** *"Modulation of midtropospheric methane by El Niño"*, https://doi.org/10.1002/2017ea000281.

- **Turner *et al.*, PNAS (2017):** *"Ambiguity in the causes for decadal trends in atmospheric methane and hydroxyl"*, https://doi.org/10.1073/pnas.1616020114.

- **Rigby *et al.*, PNAS (2017):** *"Role of atmospheric oxidation in recent methane growth"*, https://doi.org/10.1073/pnas.1616426114.

- **Turner *et al.*, PNAS (2018):** *"Modulation of hydroxyl variability by ENSO in the absence of external forcing"*, https://doi.org/10.1073/pnas.1807532115.

Regards,
Alexander J. Turner

---

## Short Comment (SC2) · 6 Sep 2018

I have short comments on Schafer et al., (2018) BGD. Generally, I think the topic of this paper is very interesting and it could be an important contribution to the community. However, I found the title is a bit misleading as it sounds like ENSO has limited impacts on CH4 sources and sinks, which is not supported by previous studies. Also, some of the assumptions used in speculating CH4 sources-related statements/conclusions are not fully justified and need to be clarified. Here below are my specific comments:

-The response of CH4 concentration to natural CH4 sources could be weak during weak/moderate ENSO events and the methane sources from anthropogenic activities could be dominating. Also, the general assumption of lower CH4 during El Ninos

[Figure]

seems to be controversial to the observations in some El Nino event (e.g. 1997/98). I feel it would be very helpful if the authors could add an additional analysis to maximize the signals by focusing on strong ENSO events.

-For C13-CH4, the authors assume that a detectable change in C13-CH4 during ENSO should be observed if ENSO has significant impacts on wetland and biomass burning given that the suppression of wetland and enhanced biomass burning act in the same direction on C13-CH4. I wonder if this signal can be detected without removing noises from other factors like atmospheric transport, local OH, and other biogenic sources which are also influenced by climate conditions (e.g. landfills and agricultural sources, which have similar C13-signature as wetlands and also respond to changing rainfall and temperature). In addition, the growth of wetland CH4 emissions during El Ninos is more complex than previous thoughts, Zhang et al., (2018) suggest that wetland CH4 emissions were suppressed at the early stage of El Nino but the wetland CH4 growth rate is in the rising phase at the later stage of El Ninos. Given that the peak of CH4 growth for wetland and biomass burning occur differently, this could weaken the net impact on the C13-CH4 signal.

- Zhang et al., (2018) suggests that wetland CH4 emission could have a step increase of $\sim$ 9 Tg CH4/yr for the period of 2007-2014 compared to 2000-2006. Could this affect some of the authors' conclusions?

Reference: Zhang Z., Zimmermann E. N., Calle L., Hurtt G., C. A., and Poulter B. 2018. Enhanced response of global wetland methane emissions to the 2015–2016 El Niño-Southern Oscillation event. Environmental Research Letters 13:074009.

---

## Author Comment (AC2) · 24 Sep 2018

We thank Joe Melton for the constructive and helpful comments and suggestions. Below we address each criticism individually. Please note that some points were brought up by several referees and commenters; please see our other responses for additional information and changes to the manuscript. Referees' comments are bracketed as follows: <>. Our response is in regular font. Quotes from the manuscript are in quotation marks.

Response to main comments:

<1. I didn't see a discussion of significance level for the correlation coefficients. Without one I have trouble understanding if a value of 0.25 is significant or not. Is there a reason

that was not done? Otherwise, while all of the r2 are low, it might help understand what are just noise and what is representing a true signal.>

We have marked all correlation results for the Spearman ranks that are not statistically robust at a level of p>0.01 with grey backgrounds in the results tables. This applies only to results with very low r-squared values and therefore does not affect the interpretation. The exception are correlations of global $\delta$13CH4 with ENSO, which all have p>0.05. These correlations did not reveal a strong ENSO impact either, so removing them from the interpretation does not alter the findings.

The resulting changes in the manuscript are as follows:

Section 5.2.: "P-values for the Spearman ranks indicate that all results for r2>0.1 are significant (p<0.001), with the exception of global $\delta$13CH4 correlations, where no p-values below 0.05 occur."

Section 5.2.: "None of the global $\delta$13CH4 series produced statistically robust correlations with ENSO; all p-values were higher than 0.05. The following findings are therefore not relevant for further interpretation. The highest correlation is between global detrended $\delta$13CH4 and SOI monthly means with r2=0.37."

Section 5.2.: "The lack of statistical robustness for global $\delta$13CH4-ENSO correlations may stem from the different resolution of the two sets of time series. In this case, the southern hemispheric record from BHD may represent the extra-tropical impact of ENSO on $\delta$13CH4."

<2. There is relatively little discussion on possible changes in the main sink of methane - hydroxyl radicals. The dynamics of this sink has been highlighted in recent studies (McNorton et al. 2016, Turner et al. 2017, Rigby et al. 2017). Is it possible that ENSO would have an impact upon local concentrations of OH? Is it a safe assumption to assume a constant sink strength? Given the power of this sink, and its recent importance (at least in the studies just mentioned) perhaps it is worthwhile to give more consideration to how a changing sink due to ENSO could impact upon the methane cycle. Or at least expand the discussion of the sink to demonstrate why a constant assumption is valid.>

A detailed discussion of OH-dynamics has been included in the revised manuscript, for details please see replies to SC1 by Alex Turner.

Response to minor comments:

<- What about using the GFED4(s) burned area product to investigate changes in burning? It would have the advantage that it is global and extends further back in time than the HCN record. However the caveat is that burned area is not the same as C emitted as CH4, however it might be a reasonable test since they would be closely related. A recent study highlighted the continual decrease in burnt area over the course of the record (Andela et al. 2017) which then should have some impact upon the methane cycle and perhaps can be used to tease out any ENSO influence.>

Evidence for an influence of ENSO on earlier versions of the GFED data set has been presented by Van der Werf et al. (2006). Although GFED now covers a longer period, we see limited value in repeating the analysis. The biomass burning proxy HCN as an atmospheric tracer provides a more direct comparison with the methane records (e.g., both are subject to atmospheric transport and mixing). Given that the focus of our study is the overall impact of ENSO on methane dynamics, with biomass burning emissions as a piece of the puzzle, we prefer to maintain a clear scope of the work.

<- Can we have a table with the various tests laid out (det-nom, det-gro, nominal mm, nominal run, etc.)? It is difficult to keep them all in the head and then interpret some very busy figures.>

We have included a new Table 1 that provides a list and description of all data sets.

<- p. 2 line 30 - What is aggregate source here?>

We have clarified that 'aggregate source' applies to the combined total of global emissions. The relevant passage now reads: Introduction: "Biogenic methanogenesis in wetlands discriminates strongly against 13C and creates methane that is 13C-depleted ($\delta$13C = -58‰ for tropical wetlands) relative to the plant precursor material ($\delta$13C of -12‰ to -28‰ and to the combined total of global emissions ($\delta$13C $\sim$ -53.5‰."

<- Author contributions - there is no E.D author.>

We have corrected the author contributions. Before the original submission we mutually agreed that Ed Dlugokencky's contribution would be more appropriately reflected in an acknowledgement rather than through co-authorship.

<- Data avail - raw data is little use. Best approach would be to make the actual analysis available.>

We are now stating that all data products and time series used in the study are available from the corresponding author.

---

## Author Comment (AC3) · 24 Sep 2018

We thank Alex Turner for the time taken to evaluate our study and for the helpful comments and suggestions. Below we address each criticism individually. Please note that some points were brought up by several referees and commenters; please see our other responses for additional information and changes to the manuscript. Referees' comments are bracketed as follows: <>. Our response is in regular font. Quotes from the manuscript are in quotation marks.

<The manuscript argues for a limited role of ENSO on the methane cycle; however, the manuscript makes little mention of two important factors that impact atmospheric methane and are strongly influenced by ENSO: (1) atmospheric transport and (2) loss

via hydroxyl. These factors seem particularly pertinent to a discussion of the role of ENSO on the methane cycle. There have been a number of recent papers on these two topics in the last two years that the authors seem to have overlooked. Specifically, McNorton et al. (2016), Turner et al. (2017), and Rigby et al. (2017) showed how changes in the methane loss via oxidation by hydroxyl was an important factor in the interpretation of methane trends.>

These papers are relevant for a complete picture of the methane cycle in the introduction and conclusion. We have added references in these sections. In contrast, for the core question of our study they are not relevant, just as anthropogenic sources like fossil-fuel methane are not. We note that a recent OH-reconstruction (Naus et al., 2018) finds that sink trends may be less relevant than modelled by Turner et al. (2017) and Rigby et al. (2017).

Changes to the manuscript are as follows:

Introduction: "Considering recent reconstructions of methane's dominant atmospheric sink, i.e. the hydroxyl radical OH, we consider it likely that increasing emissions contribute to (Rigby et al., 2017), if not dominate (Naus et al., 2018), the [CH4] rise. If so, the methane source type that varied can be investigated..."

Introduction: "Changes in OH have also been suggested as partial or dominant drivers in recent CH4 trends, both for the onset of the 1999-2006 plateau (McNorton et al., 2016; Schaefer et al., 2016) and for the post-2007 [CH4] increase (Rigby et al., 2017; Turner et al., 2017)."

Conclusions: "Changes in removal rates via OH have been suggested as an additional (Rigby et al., 2017) or alternative (Turner et al., 2017) driver of the increase, but recent work suggests that sink impacts are not dominant (Naus et al., 2018)."

<More directly related to ENSO, Corbett et al. (2017) showed the influence of ENSO on the spatial distribution of methane via changes in atmospheric transport... >

The role of transport is relevant for the tropical time series. We have included a discussion of the findings of Corbett et al. (2017) as a possible explanation why ENSO signals are smaller than may have been anticipated. However, we also note that the observed anomalies in mid-tropospheric [CH4] are inconsistent with the patterns expected from emission changes. On hemispheric or global scales transport processes are unlikely to play a strong role, given the short mixing time of methane relative to its atmospheric turn-over.

Changes to the manuscript are as follows:

Section 5.3.7.: "Corbett et al. (2017) show that during La Niña events high surface temperatures over the Western Pacific lead to upward transport over the Indonesian region (a CH4 source area from wetlands and rice paddies) and negative CH4 anomalies in the mid-troposphere (tropical surface air with relatively low [CH4] replaces air from the Northern Hemisphere with higher [CH4]). This mechanism would dampen the signal of higher La Niña emissions in surface records like SMO and ASC. However, the corresponding El Niño anomalies in mid-tropospheric CH4 over the Central Pacific are smaller. This indicates that Central Pacific surface air, where there are no CH4 sources, is closer in [CH4] to mid-tropospheric levels than surface air from the Western Pacific. Unless there were strong longitudinal differences in mid-tropospheric [CH4], this is inconsistent with a scenario where high concentrations of CH4 are generated over the Western Pacific in La Niñas but transported upwards and away from the surface stations used in this study. On hemispheric or global scales transport processes are unlikely to play a strong role, given the short mixing time of methane relative to its atmospheric turn-over."

<...while Turner et al. (2018) showed how ENSO can strongly influence the methane lifetime.>

The findings of Turner et al. (2018) on tropical OH-dynamics during ENSO events are very relevant to this study. They are now laid out briefly in the introduction and are

discussed in depth in the revised section 5.3.7. In short, OH-dynamics are expected to provide a negative feedback on methane concentration signals from ENSO emissions but a positive feedback on the stable isotope signal. This offers an additional explanation why ENSO impacts on methane growth rates is less than has been suggested in some studies. It would also make $\delta13CH4$ a more sensitive tracer for ENSO impacts but our records do no show the expected $\delta13CH4$ signals.

Changes to the manuscript that result from this discussion are as follows:

Abstract: "Dynamics of the removal by hydroxyl may counteract the variation in emissions, but the expected isotope signal is not evident."

Introduction: "A chemistry climate model suggests that ENSO modulates tropical OH (where hydroxyl levels are highest) via changes in NOx production through lightning, ozone availability and specific humidity, as well as emissions of reactive carbon (Turner et al., 2018). Resulting changes in methane removal could create their own signal in atmospheric records of [CH4] and $\delta13CH4$. They could also either reinforce or dampen the emission impacts discussed above."

Section 5.3.7.: "The low correlations of [CH4] and $\delta13CH4$ with ENSO rule out a dominant role for ENSO triggered sink changes in atmospheric methane records. Removal processes could lead to either amplification or dampening of source signals. Higher emissions of methane and CO from biomass burning will draw down OH and weaken the sink. Emission factors from fires for CO are between 10 and 30-fold higher than for CH4 (Van der Werf et al., 2017), so that the biomass burning dynamics dominate the source of reactive carbon, leaving less OH during El Niños and more during La Niñas to draw down CH4. This would provide a negative feedback for the emissions [CH4]-signal from ENSO forcing. In contrast, the feedback on the ENSO emissions $\delta13CH4$-signal would be positive due to varying enrichment of 13C-methane through sink fractionation (less removal leads to less 13C-enrichment of relatively 13C-depleted wetland emissions during La Niñas; more removal increases the 13C-enrichment from

biomass burning emissions during El Niños further). In addition to the reactive carbon effect, (Turner et al., 2018) found a further OH increase during La Niñas due to higher lightning rates with NOx production. Turner et al. (2018) could attribute 17% of OH variability that is forced by climate cycles (rather than emissions of other atmospheric compounds) to ENSO. This is a minor part of the variability, but in consequence, the dampening effect on [CH4] and the reinforcing feedback on $\delta$13CH4 would be even larger. In our correlation results these sink impacts are not apparent, as the [CH4] correlations for the tropical stations are higher than $\delta$13CH4 correlations (Tables 1 and 2). Nevertheless, the OH-dynamics provide a possible explanation for the limited ENSO impact on [CH4] variability and trends. They also make $\delta$13CH4 a conservative proxy for the influence that ENSO exerts on tropical methane. Whether ENSO has less influence on CH4 emissions than assumed or whether such an impact is overwhelmed by atmospheric removal or other CH4 cycle processes,..."

Conclusions: "As $\delta$13CH4 is subject to a mutually reinforcing signal from ENSO suppression of wetland emissions and enhancement of biomass burning CH4 (or vice versa), as well as positive feedbacks from OH-dynamics, it is particularly suited to study the role of ENSO in the CH4 cycle. Conclusions: "Counteracting OH-dynamics are expected to further dampen any influence ENSO may have on methane growth rates."

Conclusions: "Our results do not rule out that ENSO influences CH4 emissions from wetlands and biomass burning through temperature, enhanced precipitation or droughts in key regions, but any such impacts are overwhelmed by OH-dynamics or other source and sink processes."

---

## Author Response (AR1)

Dear Fortunat,

Thank you for your assistance in the publication process of our manuscript.

This is a point-by-point reply to all comments made by referees and in short comments, as well as your own. Below these, we have listed additional corrections and changes to the manuscript that became necessary during the revisions but are not directly related to external comments. A revised manuscript and a copy that shows a mark-up of all changes have also been submitted. Please note that we have added Gordon Brailsford as additional author, as discussed before.

Sincerely,

Hinrich

We thank all referees and commenters for the time taken to evaluate our study and for the helpful comments and suggestions. Below we address each criticism individually. Some points were brought up by several referees and commenters; in those cases our responses and changes to the manuscript may not be repeated for every single one, but all concerns have been addressed. Referees' comments are shown in italics. Our response is in regular font. Quotes from the manuscript are in quotation marks; bold type indicates new wording in the manuscript.

Editor's comment:

*<Personally, I missed a discussion on the potential variability in the signature of d13C(CH4) of different sources as well as of the sinks and how such variations may affect your results. For example, I could imagine that variations in the C3 versus C4 origin of organic source material could have an influence on wetland d13C(CH4) emissions.>*

Unfortunately, we had missed this comment in the acceptance letter for discussion and therefore did not address it with the earlier revisions. Variability in sinks is now been discussed in response to comments from other referees and commenters. We have now added the following passage in the introduction on the variability of source isotopic signatures and C3/C4 wetlands:

"Varying contributions from wetlands dominated by $C_3$ and $C_4$ plants, which differ in the $\delta^{13}CH_4$ of their emissions, may be part of the ENSO-CH4 signal or work to obscure it if controlled by other drivers. In general, we assume that $\delta^{13}CH_4$ of the various emission sources has not changed over the ~35 yr period of our study. Although such changes, correlated to atmospheric $CO_2$ mole fractions, have been reported to occur over centuries to millennia in ice core studies (Möller et al., 2013), they are likely negligible over the short duration and >20% $CO_2$-change of our study period."

*Referee #1:*

*<The manuscript (…) provides additional evidence that*
*emissions sources that have high interannual-variability, i.e., wetlands and fire, are*
*unlikely to be the dominant cause of sustained emissions.>*

This is an important point that we now state in the conclusions:

Conclusions: "**The longer the atmospheric [CH₄] and δ¹³CH₄ trends persist, the less probable are processes that impact IAV and short-lived cyclical events like ENSO as the driver.**"

*<First, I recommend the authors revisit the title and modify to be more specific than just 'methane cycle' because this implies the authors were looking at methane emissions, but rather the authors investigated atmospheric concentrations. I would prefer a title along the lines of "Limited impact of El Niño – Southern Oscillation on the atmospheric methane growth anomalies".>*

We have changed the title to: "**Limited impact of El Niño – Southern Oscillation on variability and atmospheric growth rate of methane**". This addresses the concerns voiced by the reviewer and also reflects that the revised manuscript also discusses sink dynamics.

*<Second, the Introduction could be clearer to reflect that the authors are motivated by understanding atmospheric methane concentration anomalies rather than anomalies in emissions. The previous studies linking methane emissions to ENSO as a key driver are not in question, but currently the Introduction mixes a little the emissions and concentrations anomalies making the reader have to work to clarify this.>*

We have clarified the scope and focus of the study. Please note that changes in response to SC2 are also relevant to this point. Resulting changes to the manuscript are as follows:

Abstract: "Here, we test the impact of ENSO on **atmospheric** CH₄ in a correlation analysis."

Introduction: "Attributing recent changes in the methane budget, **and the associated impact on its growth rate**, to specific natural or anthropogenic causes is essential for climate change mitigation."

Introduction: "We conduct correlation analyses between ENSO variability and [CH₄], as well as δ¹³CH₄ records to quantify **how much ENSO anomalies in emissions and sinks affect atmospheric CH₄**. Specifically, we explore how much of the year-to-year variability in methane **levels** can be attributed to ENSO…"

Introduction: "The aim is to detect the impact of ENSO on atmospheric CH₄ **levels** on various spatial scales."

Conclusions: "Further identification of these **processes** is necessary to inform climate change mitigation policies and climate projections."

Further, following suggestions from both reviewers and an additional short comment, the revised manuscript also discusses sink dynamics. The relationship between the latter and variability in emissions is now clarified in the introduction. For more details on the treatment of sink dynamics please see the response to SC1.

*<In Table 1, I assume the lag time is in months, so 54 is a 54 month lag? If so, many are longer than 12 months, which is contrary to the statement in Section 5.2 that says most are shorter than a year.>*

We have clarified that lag times in the tables are reported in months, e.g. by including this information in the table captions.

The statement that detrended time series at SMO generally have their highest correlations at lag times of less than one year

holds true (the only exceptions are for EMI, as well as a few cases for SMO det-gro). We have modified the text to make it clearer that the short lag times only apply to SMO detrended data series.

Section 5.1.2.: "…our analysis therefore allows for lag times of up to 5 years in monthly increments in the calculations and reports the maximum $r^2$ and lag time **(in months)** for a given ENSO-[CH$_4$]/$\delta^{13}$CH$_4$/HCN combination."

5 Section 5.2.: "Methane mixing ratios show correlations with ENSO of $r^2$-values up to 0.36 at SMO, but only for detrended time series (Table 1). […] **For SMO detrended [CH$_4$] series**, lag times are fairly consistent across the various ENSO indices and generally shorter than 1 year. For other [CH$_4$] records at SMO and ASC the **highest** correlations are $r^2$<0.23 **and have** lags of over 3 years.

10 *<I am skeptical of such long lags, it is difficult to judge whether*
*shorter lags were close in terms of significance to the longer lag times because these*
*numbers are not presented.>*

We agree that allowing for lags up to 5 years leads to questionable results. However, it is difficult to define a cut-off for lags. The current presentation provides an upper limit for ENSO influence. Given that the latter is found to be low, this represents

15 a conservative estimate. For most dependent time series, there are cases of comparable r2-values for lags both longer and shorter than 3 years. Using a 3-year cut-off for lags therefore does not really affect the conclusions. Nevertheless, for specific r2-values with lags over 3 years that are mentioned in results and discussion, we now also report the corresponding highest r2 for lags of less than 3 years. We have revised sections 5.1.2. and 5.2. to address this point.

Section 5.1.2.: "A lag time between ENSO forcing and detection of resulting $\delta^{13}$CH$_4$ or HCN variability at the measurement

20 site, (or in the global average) is likely, due to a **variety of factors that may lead to lags of unknown length and some of which may be cumulative: e.g., hydrology, plant growth and decay, microbial response, seasonal triggers for methanogenesis or burning, as well as atmospheric chemistry, mixing, and transport between source regions and sampling sites. Therefore, it is difficult to define a cut-off for lags.** Literature estimates of **specific** lags range…"

Section 5.2.: "**Although the analysis provides $r^2$-values for lags up to 60 months (Tables 2-4), we consider it likely that**

25 **lags of >3 years indicate spurious correlations given that individual ENSO events last 1-2 years and global atmospheric mixing times are on order of 1 year. Therefore, we also report the highest $r^2$ for lags <3 years in the following sections. For other cases with lags >3 years in Tables 2-4, the highest relevant $r^2$-value is lower than the reported value, where the latter places an upper limit on the influence of ENSO**."

Section 5.2.: "For other [CH$_4$] records at SMO and ASC the highest correlations are $r^2$<0.23 and have lags of over 3 years

30 (**$r^2$<0.19 for lags <3 years**). The global running mean [CH$_4$] time series shows $r^2$=0.24 (lag: 4.5 years**; $r^2$=0.04 for lag <3 years**) with the SOI running mean for the period 1998-2016.

*Referee #2, Joe Melton:*

We thank Joe Melton for the constructive and helpful comments and suggestions. Below we address each criticism individually. Please note that some points were brought up by several referees and commenters; please see our other responses for additional information and changes to the manuscript. Referees' comments are bracketed as follows: <>. Our response is in regular font. Quotes from the manuscript are in quotation marks.

Response to main comments:

*<1. I didn't see a discussion of significance level for the correlation coefficients. Without one I have trouble understanding if a value of 0.25 is significant or not. Is there a reason that was not done? Otherwise, while all of the r2 are low, it might help understand what are just noise and what is representing a true signal.>*

We have marked all correlation results for the Spearman ranks that are not statistically robust at a level of p>0.01 with grey backgrounds in the results tables. This applies only to results with very low r-squared values and therefore does not affect the interpretation. The exception are correlations of global $\delta13CH_4$ with ENSO, which all have p>0.05. These correlations did not reveal a strong ENSO impact either, so removing them from the interpretation does not alter the findings. The highest correlations have been revised downwards by a slight margin.

The resulting changes in the manuscript are as follows:

Abstract: "We find that at most **36%** of the variability in [CH$_4$] and $\delta^{13}CH_4$ is attributable to ENSO, but only for detrended records in the Southern tropics. Trend-bearing records from the Southern tropics, as well as all studied hemispheric and global records show a minor impact of ENSO, i.e. **<24%** of variability explained."

Section 5.2.: "**P-values for the Spearman ranks indicate that all results for $r^2$>0.1 are significant (p<0.001), with the exception of global $\delta^{13}CH_4$ correlations, where no p-values below 0.05 occur**."

Section 5.2.: "**None of the global $\delta^{13}CH_4$ series produced statistically robust correlations with ENSO; all p-values were higher than 0.05. The following findings are therefore not relevant for further interpretation**. The highest correlation is between global detrended $\delta^{13}CH_4$ and SOI monthly means with $r^2$=0.37."

Section 5.2.: "**The lack of statistical robustness for global $\delta^{13}CH_4$-ENSO correlations may stem from the different resolution of the two sets of time series. In this case, the southern hemispheric record from BHD may represent the extra-tropical impact of ENSO on $\delta^{13}CH_4$.**"

*<2. There is relatively little discussion on possible changes in the main sink of methane - hydroxyl radicals. The dynamics of this sink has been highlighted in recent studies (McNorton et al. 2016, Turner et al. 2017, Rigby et al. 2017). Is it possible that ENSO would have an impact upon local concentrations of OH? Is it a safe assumption to assume a constant sink strength? Given the power of this sink, and its recent importance (at least in the studies just mentioned) perhaps it is worthwhile to give more consideration to how a changing sink due to ENSO could impact upon the methane cycle. Or at least expand the discussion of the sink to demonstrate why a constant assumption is valid.>*

A detailed discussion of OH-dynamics has been included in the revised manuscript, for details please see replies to SC1 by Alex Turner.

Response to minor comments:

*<- What about using the GFED4(s) burned area product to investigate changes in burning? It would have the advantage that it is global and extends further back in time than the HCN record. However the caveat is that burned area is not the same as C emitted as CH4, however it might be a reasonable test since they would be closely related. A recent study highlighted the continual decrease in burnt area over the course of the record (Andela et al. 2017) which then should have some impact upon the methane cycle and perhaps can be used to tease out any ENSO influence.>*

Evidence for an influence of ENSO on earlier versions of the GFED data set has been presented by Van der Werf et al. (2006). Although GFED now covers a longer period, we see limited value in repeating the analysis. The biomass burning proxy HCN as an atmospheric tracer provides a more direct comparison with the methane records (e.g., both are subject to atmospheric transport and mixing). Given that the focus of our study is the overall impact of ENSO on methane dynamics, with biomass burning emissions as a piece of the puzzle, we prefer to maintain a clear scope of the work.

*<- Can we have a table with the various tests laid out (det-nom, det-gro, nominal mm, nominal run, etc.)? It is difficult to keep them all in the head and then interpret some very busy figures.>*

We have included a new Table 1 that provides a list and description of all data sets.

*<- p. 2 line 30 - What is aggregate source here?>*

We have changed the wording to read:

Introduction: "Biogenic methanogenesis in wetlands discriminates strongly against $^{13}$C and creates methane that is $^{13}$C-depleted ($\delta^{13}$C = -58‰ for tropical wetlands) relative to the plant precursor material ($\delta^{13}$C of -12‰ to -28‰) and to the **combined total of global emissions** ($\delta^{13}$C ~ -53.5‰)."

*<- Author contributions - there is no E.D author.>*

We have corrected the author contributions. Before the original submission we mutually agreed that Ed Dlugokencky's contribution would be more appropriately reflected in an acknowledgement rather than through co-authorship.

*<- Data avail - raw data is little use. Best approach would be to make the actual analysis available.>*

We are now stating that all data products and time series used in the study are available from the corresponding author.

*Interactive comment SC1:*

We thank Alex Turner for the time taken to evaluate our study and for the helpful comments and suggestions. Below we address each criticism individually. Please note that some points were brought up by several referees and commenters; please see our other responses for additional information and changes to the manuscript. Referees' comments are bracketed as follows: <>. Our response is in regular font. Quotes from the manuscript are in quotation marks.

*<The manuscript argues for a limited role of ENSO on the methane cycle; however, the manuscript makes little mention of two important factors that impact atmospheric methane and are strongly influenced by ENSO: (1) atmospheric transport and (2) loss via hydroxyl. These factors seem particularly pertinent to a discussion of the*
5    *role of ENSO on the methane cycle. There have been a number of recent papers on these two topics in the last two years that the authors seem to have overlooked. Specifically, McNorton et al. (2016), Turner et al. (2017), and Rigby et al. (2017) showed how changes in the methane loss via oxidation by hydroxyl was an important factor in the interpretation of methane trends.>*

These papers are relevant for a complete picture of the methane cycle in the introduction and conclusion. We have added references in these sections. In contrast, for the core question of our study they are not relevant, just as anthropogenic sources like fossil-fuel methane are not. We note that a recent OH-reconstruction (Naus et al., 2018) finds that sink trends may be less relevant than modelled by Turner et al. (2017) and Rigby et al. (2017).

 Changes to the manuscript are as follows:

Introduction: "**Considering recent reconstructions of methane's dominant atmospheric sink, i.e. the hydroxyl radical OH, we consider it likely that increasing emissions contribute to (Rigby et al., 2017), if not dominate (Naus et al., 2018), the [CH₄] rise. If so,** the methane source type that varied can be investigated…"

20    Introduction: "**Changes in OH have also been suggested as partial or dominant drivers in recent CH₄ trends, both for the onset of the 1999-2006 plateau (McNorton et al., 2016; Schaefer et al., 2016) and for the post-2007 [CH₄] increase (Rigby et al., 2017; Turner et al., 2017)**."

Conclusions: "**Changes in removal rates via OH have been suggested as an additional (Rigby et al., 2017) or alternative (Turner et al., 2017) driver of the increase, but recent work suggests that sink impacts are not dominant (Naus et al.,**
25    **2018)**."

*<More directly related to ENSO, Corbett et al. (2017) showed the influence of ENSO on the spatial distribution of methane via changes in atmospheric transport… >*

30    The role of transport is relevant for the tropical time series. We have included a discussion of the findings of Corbett et al. (2017) as a possible explanation why ENSO signals are smaller than may have been anticipated. However, we also note that the observed anomalies in mid-tropospheric [CH4] are inconsistent with the patterns expected from emission changes. On hemispheric or global scales transport processes are unlikely to play a strong role, given the short mixing time of methane relative to its atmospheric turn-over.

35    Changes to the manuscript are as follows:

Section 5.3.7.: "**Corbett et al. (2017) show that during La Niña events high surface temperatures over the Western Pacific lead to upward transport over the Indonesian region (a CH₄ source area from wetlands and rice paddies) and negative CH₄ anomalies in the mid-troposphere (tropical surface air with relatively low [CH₄] replaces air from the Northern Hemisphere with higher [CH₄]). This mechanism would dampen the signal of higher La Niña emissions in**

**surface records like SMO and ASC. However, the corresponding El Niño anomalies in mid-tropospheric CH₄ over the Central Pacific are smaller. This indicates that Central Pacific surface air, where there are no CH₄ sources, is closer in [CH₄] to mid-tropospheric levels than surface air from the Western Pacific. Unless there were strong longitudinal differences in mid-tropospheric [CH₄], this is inconsistent with a scenario where high concentrations of CH₄ are**

5  **generated over the Western Pacific in La Niñas but transported upwards and away from the surface stations used in this study. On hemispheric or global scales transport processes are unlikely to play a strong role, given the short mixing time of methane relative to its atmospheric turn-over**."

*<…while Turner et al. (2018) showed how ENSO can strongly influence the methane lifetime.>*

10  The findings of Turner et al. (2018) on tropical OH-dynamics during ENSO events are very relevant to this study. They are now laid out briefly in the introduction and are discussed in depth in the revised section 5.3.7. In short, OH-dynamics are expected to provide a negative feedback on methane concentration signals from ENSO emissions but a positive feedback on the stable isotope signal. This offers an additional explanation why ENSO impacts on methane growth rates is less than has been suggested in some studies. It would also make δ13CH4 a more sensitive tracer for ENSO impacts but our records do no

15  show the expected δ13CH4 signals.

Changes to the manuscript that result from this discussion are as follows:

Abstract: "**Dynamics of the removal by hydroxyl may counteract the variation in emissions, but the expected isotope signal is not evident.**"

Introduction: "**A chemistry climate model suggests that ENSO modulates tropical OH (where hydroxyl levels are**

20  **highest) via changes in NOx production through lightning, ozone availability and specific humidity, as well as emissions of reactive carbon (Turner et al., 2018). Resulting changes in methane removal could create their own signal in atmospheric records of [CH₄] and δ¹³CH₄. They could also either reinforce or dampen the emission impacts discussed above.**"

Section 5.3.7.: "**The low correlations of [CH₄] and δ¹³CH₄ with ENSO rule out a dominant role for ENSO triggered sink**

25  **changes in atmospheric methane records. Removal processes could lead to either amplification or dampening of source signals. Higher emissions of methane and CO from biomass burning will draw down OH and weaken the sink. Emission factors from fires for CO are between 10 and 30-fold higher than for CH₄ (Van der Werf et al., 2017), so that the biomass burning dynamics dominate the source of reactive carbon, leaving less OH during El Niños and more during La Niñas to draw down CH₄. This would provide a negative feedback for the emissions [CH₄]-signal from ENSO**

30  **forcing. In contrast, the feedback on the ENSO emissions δ¹³CH₄-signal would be positive due to varying enrichment of ¹³C-methane through sink fractionation (less removal leads to less ¹³C-enrichment of relatively ¹³C-depleted wetland emissions during La Niñas; more removal increases the ¹³C-enrichment from biomass burning emissions during El Niños further). In addition to the reactive carbon effect, (Turner et al., 2018) found a further OH increase during La Niñas due to higher lightning rates with NOx production. Turner et al. (2018) could attribute 17% of OH variability**

that is forced by climate cycles (rather than emissions of other atmospheric compounds) to ENSO. **This is a minor part of the variability, but in consequence, the dampening effect on [CH4] and the reinforcing feedback on $\delta^{13}CH_4$ would be even larger. In our correlation results these sink impacts are not apparent, as the [CH4] correlations for the tropical stations are higher than $\delta^{13}CH_4$ correlations (Tables 1 and 2). Nevertheless, the OH-dynamics provide a possible**

5    **explanation for the limited ENSO impact on [CH4] variability and trends. They also make $\delta^{13}CH_4$ a conservative proxy for the influence that ENSO exerts on tropical methane**. Whether ENSO has less influence on $CH_4$ emissions than assumed or whether such an impact is overwhelmed by **atmospheric removal or** other $CH_4$ cycle processes,…"

Conclusions: "As $\delta^{13}CH_4$ is subject to a mutually reinforcing signal from ENSO suppression of wetland emissions and enhancement of biomass burning $CH_4$ (or vice versa), **as well as positive feedbacks from OH-dynamics**, it is particularly

10    suited to study the role of ENSO in the $CH_4$ cycle.

Conclusions: "**Counteracting OH-dynamics are expected to further dampen any influence ENSO may have on methane growth rates**."

Conclusions: "Our results do not rule out that ENSO influences $CH_4$ emissions from wetlands and biomass burning through temperature, enhanced precipitation or droughts in key regions, but any such impacts are overwhelmed **by OH-dynamics or**

15    other source and sink processes."

*Interactive comment SC2:*

20    We thank Zhen Zhang for the time taken to evaluate our study and for the helpful comments and suggestions. Below we address each criticism individually. Please note that some points were brought up by several referees and commenters; please see our other responses for additional information and changes to the manuscript. Referees' comments are bracketed as follows: <>. Our response is in regular font. Quotes from the manuscript are in quotation marks.

25    *<I found the title is a bit misleading as it sounds like ENSO has limited impacts on CH4 sources and sinks, which is not supported by previous studies.>*

We have clarified the goal and scope of the study in the introduction. We have also changed the title following suggestions from this comment and other referees. Relevant changes to the title and manuscript are as follows:

Title: "Limited impact of El Niño – Southern Oscillation on **variability and growth rate of atmospheric methane**"

Abstract: "Here, we test the impact of ENSO on **atmospheric** $CH_4$ in a correlation analysis."

Abstract: "It is possible that other processes obscure the ENSO signal, which itself indicates a minor influence of the latter on **global** $CH_4$ emissions."

35    Introduction: "We conduct correlation analyses between ENSO variability and [CH4], as well as $\delta^{13}CH_4$ records to quantify **how much** ENSO **anomalies in emissions and sinks affect atmospheric CH4**. Specifically, we explore how much of the

year-to-year variability in methane **levels** can be attributed to ENSO and how large the ENSO-$CH_4$ signal is in dependence of latitude."

Conclusions: "Further identification of these **processes** is necessary to inform climate change mitigation policies and climate projections."

*<-The response of CH4 concentration to natural CH4 sources could be weak during weak/moderate ENSO events and the methane sources from anthropogenic activities could be dominating.>*

The goal of our study is to find out whether the cumulative effect of ENSO events on the methane cycle is strong enough to drive observed trends in atmospheric methane (see response above). This does indeed mean that other processes may dominate. In fact, this is the main finding of the paper and has been discussed explicitly in the original submission, e.g., in Section 5.3.7., which is titled "Role of other methane cycle processes" and in the conclusions. Relevant passages include: "…or other processes in the $CH_4$-cycle obscure the ENSO impacts."; "ENSO could affect $CH_4$ emissions from tropical wetlands and biomass burning as predicted by Hodson et al. (2011) and van der Werf et al. (2006), respectively, but the resulting isotopic signal is overwhelmed by other components of the $CH_4$ cycle. Such influences could be other sources (particularly anthropogenic ones), variability in atmospheric transport or changes in $CH_4$ sink processes."; and "Our results do not rule out that ENSO influences $CH_4$ emissions from wetlands and biomass burning through temperature, enhanced precipitation or droughts in key regions, but any such impacts are overwhelmed **by OH-dynamics or** other source and sink processes."

*<Also, the general assumption of lower CH4 during El Ninos seems to be controversial to the observations in some El Nino event (e.g. 1997/98).>*

The focus of our study is if the impact of ENSO on atmospheric CH4 is persistent throughout our records. Working with general assumption of lower CH4 during El Ninos, consistent with the findings in the latest paper authored by the commenter, is therefore a valid approach. The variance in expression between different ENSO events is discussed in the original submission, section 5.3.4. concluding with the sentence: "Depending on the strength and geographical expression of the climate anomaly, ENSO may thus cause regional or global emission anomalies that are opposite to the expected pattern."

*<I feel it would be very helpful if the authors could add an additional analysis to maximize the signals by focusing on strong ENSO events.>*

Attribution of signals to specific ENSO events is better approached with different methods and data sets than used in our study. Several recent publications on this topic are cited in the original submission, e.g., Pandey et al. (2017). It is also not the question we are trying to answer. Strong events may cause larger signals, but they are also rarer and more sporadic, so that their impact on our time series is limited. This is discussed in section 5.3.3. Another point on this suggestion is that the commenter's last publication identifies only four strong El Ninos, two of which occur before the start of our tropical d13CH4 and HCN records. It is impossible to find general patterns with such a small sample size. We therefore prefer to maintain the scope of our study. Please note that the original submission already provided the following finding on strong El Ninos and their impact on the correlations:

"The full BHD record for 1992-2016 gives very similar results as the 1998-2016 subset used for comparison with the other stations (as discussed above). However, the shorter subset for 1998-2014 produces larger Pearson $r^2$-values (0.26 for running means and SOI), and for 2001-2014 we find Pearson $r^2$-values up to 0.38 (growth rate correlated to EMI). These shorter data sets omit the strong El Niño events of 1998 and/or 2015-16, which could have been expected to have a strong influence on methane emissions and consequently $\delta^{13}CH_4$."

*<-For C13-CH4, the authors assume that a detectable change in C13-CH4 during ENSO*
*should be observed if ENSO has significant impacts on wetland and biomass burning*
5 *given that the suppression of wetland and enhanced biomass burning act in the same*
*direction on C13-CH4. I wonder if this signal can be detected without removing noises*
*from other factors like atmospheric transport, local OH, and other biogenic sources*
*which are also influenced by climate conditions (e.g. landfills and agricultural sources,*
*which have similar C13-signature as wetlands and also respond to changing rainfall*
10 *and temperature).>*

This comment goes back to the one above on interfering signals from other processes (see response above). While the individual arguments are all correct, they do not invalidate our point from the original submission: "If ENSO is invoked as a main cause of recent trends in [CH$_4$] and $\delta^{13}$CH$_4$ this should be manifested in sizeable correlations." We have added a note in
15 the introduction regarding the commenter's point that some anthropogenic sources could reinforce the wetland anomalies:
Introduction: "**Several anthropogenic sources are subject to the same ENSO forcing and are expected to vary in concert with wetlands (e.g., rice agriculture, possibly livestock**)."

*<In addition, the growth of wetland CH4 emissions during El Ninos is more complex than previous thoughts, Zhang et al.,*
20 *(2018) suggest that wetland CH4 emissions were suppressed at the early stage of El Nino but the wetland CH4 growth rate is*
*in the rising phase at the later stage of El Ninos.>*

It is unclear how much mixing and transport would smooth out the evolving signal from an El Nino event in our atmospheric records. Nevertheless, the sequence has the potential to cause patterns in monthly growth rates (strong positive anomalies) that
25 are in contradiction with the general pattern of reduced overall emissions. We have noted this in sections 5.2. and 5.3.4.:

"**The highest values are from (detrended) growth rates, which can be more indicative of dynamics within an ENSO event, rather than its overall emissions impact (Zhang et al., 2018).**"

30 "**Zhang et al., (2018) report an evolving response of wetland emissions to El Niños, where an initial reduction due to decreased wetland extend is counteracted by increased microbial activity under higher temperatures during the later stages of the event.**"

35 *<Given that the peak of CH4 growth for wetland and biomass burning occur differently, this could weaken the net impact on*
*the C13-CH4 signal.>*

This was explicitly stated in the original submission (Section 5.3.1.):
"It is possible that biomass burning and wetland CH$_4$ production have different response times to ENSO forcing, which would
40 weaken their cumulative impact on $\delta^{13}$CH$_4$."

*<- Zhang et al., (2018) suggests that wetland CH4 emission could have a step increase*
*of ~9 Tg CH4/yr for the period of 2007-2014 compared to 2000-2006. Could this affect*
*some of the authors' conclusions?>*

45 The finding indeed affects certain conclusions, although with the caveat that Melton et al. (2014) show that different wetland

models show a large range in modelled emissions and do not necessarily agree in the simulated trends. Agreement of the model

ensemble would be needed for a robust conclusion. We have added the following passages:

Section 5.3.6.: "**Other wetland variability may have contributed to the rise (Zhang et al., 2018); given the range in wetland model output (Melton et al., 2014) this stands to be confirmed by ensemble runs.**"

Section 6.: "Our results **do** not rule out that wetland production is a contributor to the post-2007 [CH$_4$]-rise if driven by environmental controls other than ENSO. **This is suggested by a modelled increase in wetland CH$_4$ production between the periods 2000-2006 and 2006-2014, although with the limited confidence of a single wetland emissions model (Zhang et al., 2018).**"

Please note that we have added additional passages - or amended existing ones - informed by the study of Zhang et al. (2018) with regards to various points:

Section 5.3.2.: "One explanation for the lower combined wetland-pyrogenic $\delta^{13}$CH$_4$ signal is low sensitivity of wetland CH$_4$ production to ENSO events. **This is consistent with r$^2$-values of 0.12-0.26 between modelled wetland methane emissions (using different climate data sets as drivers) and MEI as reported by Zhang et al., (2018).**"

Section 5.3.3.: "Previous findings that modelled tropical (Zhu et al., 2015) **and global (Zhang et al., 2018)** wetland CH$_4$ emissions can explain at most 25% **and 14%, respectively**, of the variation in atmospheric methane growth rates therefore agree with our results that ENSO exerts only a minor control on **atmospheric** CH$_4$." (note that the previous version read "… ENSO exerts only a minor control on global CH$_4$ emissions")

Section 5.3.4.: **(Zhang et al., 2018) report an evolving response of wetland emissions to El Niños, where an initial reduction due to decreased wetland extend is counteracted by increased microbial activity under higher temperatures during the later stages of the event.** A complex response of wetland CH$_4$ production is not only seen in models, however.

Conclusions: "There is evidence for additional methane emissions from agriculture (**Wolf et al., 2017**) …"

Additional changes to the manuscript:

Gordon Brailsford (NIWA) has been included in the author list to reflect his contributions to the BHD measurements. The author contributions have been updated accordingly.

We have added an acknowledgement to D. Lowe, who started the BHD time series.

We have corrected throughout the manuscript an error that [CH4] is reported as mixing ratio, while it is mole fraction.

We have included additional references on technical aspects of the [CH4] measurements at BHD:

"In addition, we use data measured at the NZ National Institute of Water and Atmospheric Research (NIWA) from BHD in NZ (41.41$^0$S, 174.87$^0$E; 1992-2017) **(Lowe et al., 1991). Both data sets are on the same international scale (Dlugokencky et al., 2005), although for the presented analysis internal consistency of the time series is the relevant criterion; interlaboratory offsets do not affect the findings**."

We have corrected a sentence in the introduction where the terms "enriched" and depleted" had been mixed up:

[revised manuscript text omitted]
 | 0.06 | 1 | 0.06 | 2 | 0.07 | 2 | 0.08 | 3 | 0.05 | 15 | 0.07 | 16 | 0.14 | 16 | 0.19 | 20 |